# A General Neural Backbone for Mixed-Integer Linear Optimization via Dual Attention

Peixin Huang [1]   Yaoxin Wu [2]   Yining Ma [3]   Cathy Wu [3]   Wei Zhang [4]   Wen Song [1]

## Abstract

Mixed-integer linear programming (MILP) is a foundational framework for combinatorial optimization across science and engineering, but remains hard to solve at scale due to NP-hardness. Recent learning-based methods typically model MILP instances as variable–constraint bipartite graphs and use Graph Neural Networks (GNNs) for representation learning, yet their locality limits representation power. We propose an attention-driven neural backbone that adopts an element-centric view of variables and constraints, with dual attention performing parallel intra-type self-attention and inter-type cross-attention. Across three representative tasks at the instance, element, and solving-state levels, our model consistently outperforms conventional GNN-based architectures, highlighting attention-based, element-centric modeling as a powerful foundation for learning-enhanced combinatorial optimization.

## 1. Introduction

Combinatorial Optimization Problems (COPs) are ubiquitous in science and engineering (Kalinin et al., 2025; Romera-Paredes et al., 2024; Du et al., 2024; Naseri & Koffas, 2020; Mirhoseini et al., 2021). However, a major computational bottleneck for COP solving is the well-known NP-hardness, making it very challenging to solve in practice. As a representative, Mixed-Integer Linear Programming (MILP) is a central and widely used general modeling framework for combinatorial optimization (Bengio et al., 2021; Li et al., 2025). A MILP optimizes a linear objective subject to linear constraints, where a subset of decision variables

are restricted to take integer values. While performance of modern MILP solvers such as Gurobi, CPLEX, and SCIP (Clautiaux & Ljubic, 2025) has been continuously improved during the past decades, efficiently solving large-scale practical problems remains extremely challenging due to the strong NP-hardness. Motivated by the recent success of Neural Combinatorial Optimization (NCO), deep learning has been applied in various ways to speed up MILP solvers (Scavuzzo et al., 2024), such as learning primal heuristics to find high-quality feasible solutions quickly (Ding et al., 2020; Nair et al., 2020; Khalil et al., 2022; Han et al., 2023; Huang et al., 2024; Liu et al., 2025; Song et al., 2020; Wu et al., 2021; Huang et al., 2023; Geng et al., 2025), learning branching heuristics to reduce search tree size (Gasse et al., 2019; Gupta et al., 2020; Zarpellon et al., 2021; Gupta et al., 2022; Scavuzzo et al., 2022; Zhang et al., 2024; Sun et al., 2024; Feng & Yang, 2025), learning cutting plane control policies to speed up linear programming solving at each node (Tang et al., 2020; Wang et al., 2024; Paulus et al., 2022; Ling et al., 2024; Puigdemont et al., 2024).

While deep learning for MILP is thriving, research on the backbone neural architecture is surprisingly sparse, which is the key bottleneck in this emerging field. As the foundation of learning-based MILP models, the backbone neural architecture is responsible for extracting high-level feature embeddings from raw instance data to facilitate downstream learning tasks. The mainstream and state-of-the-art approach is to model MILP instances as variable-constraint bipartite graphs, and apply Graph Neural Network (GNN) to extract variable and constraint embeddings through message passing and aggregation among neighboring nodes (Gasse et al., 2019; Cappart et al., 2023). This neural architecture encodes the structural information of MILP through local message passing, is naturally size-invariant, and is able to handle instances of varying scales.

However, the above graph-based scheme has limited power in MILP representation, mainly due to its local-oriented mechanism. Specifically, in one layer of GNN, each node can only fuse information within its one-hop neighborhood, resulting in a very limited receptive field. Stacking multiple GNN layers cannot effectively alleviate this bottleneck because node embeddings quickly become indistinguishable due to GNN's inherent over-smoothing and over-squashing

---

[1] Shenzhen Research Institute, Shandong University, China [2] Department of Industrial Engineering and Innovation Sciences, Eindhoven University of Technology, Netherlands [3] Massachusetts Institute of Technology, USA [4] School of Control Science and Engineering, Shandong University, China. Correspondence to: Wen Song <wensong@email.sdu.edu.cn>.

*Proceedings of the 43rd International Conference on Machine Learning*, Seoul, South Korea. PMLR 306, 2026. Copyright 2026 by the author(s).

issues (Li et al., 2018; Topping et al., 2022). The consequence is that each node's information source is within a small local range (normally within 4 hops in the bipartite graph), which significantly impairs the capability of graph-based architectures in capturing deeper representations and long-range dependencies that are crucial for efficiently solving general COPs. The limited expressive power of GNN in representing MILP has been noticed and theoretically studied in several works (Chen et al., 2023; 2024; 2025). However, they are still under the local-oriented framework of GNN, and essentially cannot overcome this limitation.

This paper addresses the fundamental challenge of designing a more powerful backbone representation learning architecture for MILP. Different from existing works, we propose to view an MILP instance as two types of heterogeneous elements, i.e., variables and constraints. Then, we design a novel neural architecture to extract element embeddings through a dual-attention mechanism, as illustrated in Fig. 1. The first aspect is to analyze the intra-type relationships through full self-attention, enabling each element to directly access information from any other element of the same type. The second aspect is to capture inter-type relationships through bidirectional cross-attention, so as to exploit variable-constraint dependencies inherent in the MILP instance's structure. With this design, each element can globally establish direct and adaptive connections with long-range ones, significantly enhancing its receptive field. Moreover, more layers can be stacked to enhance the quality of generated embeddings, thereby overcoming limitations of existing graph-based architectures.

Through this paradigm, we aim to establish a new foundation for deep learning-based general combinatorial optimization solving. Our architecture serves as a general backbone that can be readily applied to diverse MILP learning tasks, including instance-level prediction (e.g., feasibility and optimal objective), element-level prediction (e.g., optimal solution), and solving state-level prediction (e.g., branching policy). Extensive experiments on problems from various domains demonstrate that the proposed attention-based architecture exhibits substantially stronger expressive power than existing GNN counterparts, suggesting that attention-driven representations may constitute a new paradigm for learning to solve general COPs.

## 2. Preliminaries

**Mixed-Integer Linear Programming.** A MILP with $n$ variables and $m$ constraints can be formally defined as

$$\min \mathbf{c}^\top \mathbf{x},$$
$$\text{s.t.} \quad \mathbf{A}\mathbf{x} \leq \mathbf{b}, \quad \mathbf{l} \leq \mathbf{x} \leq \mathbf{u}, \quad \mathbf{x} \in \mathbb{Z}^q \times \mathbb{R}^{n-q} \quad (1)$$

where $\mathbf{x}$ is a vector of $n$ decision variables, with $q$ of them being integer variables and the rest $n-q$ variables being continuous. $\mathbf{c} \in \mathbb{R}^n$ represents the coefficients of each variable in the objective function. $\mathbf{A} \in \mathbb{R}^{m \times n}$ is the coefficient matrix of the $m$ constraints, and $\mathbf{b} \in \mathbb{R}^m$ is the corresponding right-hand side values. $\mathbf{l} \in (\mathbb{R} \cup \{-\infty\})^n$ and $\mathbf{u} \in (\mathbb{R} \cup \{+\infty\})^n$ denote the lower and upper bounds for each variable, respectively. MILP is very general and broadly applicable to many COPs.

**Graph-based MILP Representation.** The key of learning-based MILP solving is to extract *embeddings* of variables to enable downstream tasks. To this end, (Gasse et al., 2019) proposed a two-stage GNN that has become the de facto architecture for MILP representation learning. It models an MILP instance as a bipartite graph $\mathcal{G} = (\mathcal{V}, \mathcal{C}, \mathcal{E})$, where $\mathcal{V} = \{v_1, v_2, ..., v_n\}$ is the variable node set with $v_i$ representing $i$-th variable, $\mathcal{C} = \{c_1, c_2, ..., c_m\}$ is the constraint node set with $c_j$ representing the $j$-th constraint, and $\mathcal{E} = \{e_{ij} | \mathbf{A}_{ij} \neq 0, \ \forall 1 \leq i \leq n, 1 \leq j \leq m\}$ is the edge set with $e_{ij}$ representing a nonzero constraint coefficient in the MILP instance. Let $\mathbf{B} \in \{0,1\}^{n \times m}$ be the bipartite adjacency matrix with $\mathbf{B}_{ij} = 1$ if $e_{ij} \in \mathcal{E}$. Each element $v_i \in \mathcal{V}$, $c_j \in \mathcal{C}$ and $e_{ij} \in \mathcal{E}$ is equipped with a raw feature vector $h_i^V \in \mathbf{H}_0^V$, $h_j^C \in \mathbf{H}_0^C$ and $h_{ij}^E \in \mathbf{H}_0^E$ to describe numerical and structural information of the MILP instance (listed in Appendix B). These raw features are first mapped to a $d$-dimensional space, and then pass through a two-stage bipartite GNN (BGNN) process:

$$\mathbf{H}^C \leftarrow \text{MLP}^C \left( \mathbf{H}^C, (\mathbf{B}^\top \mathbf{H}^V, \mathbf{H}^E) \right),$$
$$\mathbf{H}^V \leftarrow \text{MLP}^V \left( \mathbf{H}^V, (\mathbf{B}\mathbf{H}^C, \mathbf{H}^E) \right) \quad (2)$$

where $\mathbf{H}^C = \text{MLP}_0^C(\mathbf{H}_0^C) \in \mathbb{R}^{m \times d}$ and $\mathbf{H}^V = \text{MLP}_0^V(\mathbf{H}_0^V) \in \mathbb{R}^{n \times d}$ denote the embeddings of constraints and variables, $\mathbf{H}^E = \mathbf{H}_0^E W_0^E \in \mathbb{R}^{|\mathcal{E}| \times d}$ denotes edge feature embeddings, $\text{MLP}_0^V$, $\text{MLP}_0^C$, $\text{MLP}^C$ and $\text{MLP}^V$ are multi-layer perceptrons with ReLU activation, $W_0^E$ is a learnable parameter. Essentially, each constraint first receives messages from its immediate variables to update its own embedding, and then each variable performs similar operations considering its immediate constraints.

In this graph-based scheme, message passing occurs only within the one-hop neighborhood of each variable or constraint node. In principle, accessing information from a node that is $k$ hops away requires $k$ rounds of message passing, i.e., $\lceil k/2 \rceil$ BGNN layers. However, this is impractical for long-range nodes because the discriminative power of BGNN rapidly deteriorates with the increase of layers (which will be demonstrated in the experiments). Consequently, most existing works employ only 1 or 2 BGNN layers. This inherent limitation restricts the model's ability in extracting deeper representations and capturing long-range dependencies between variable and constraint nodes.

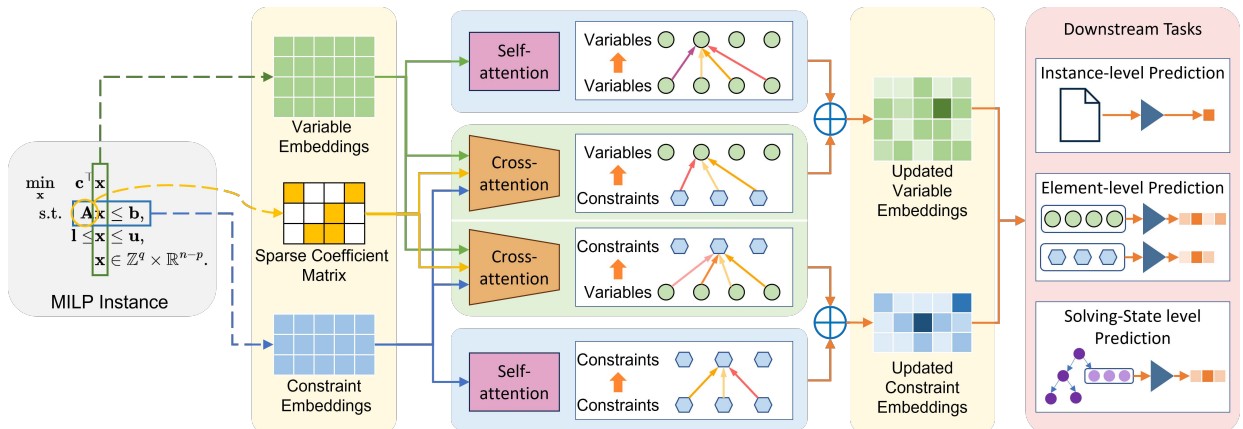

*Figure 1.* **Overview of the proposed attention-driven architecture.**

## 3. Method

In this paper, we go beyond the purely graph-centric view of MILP by treating variables and constraints as two types of key elements in MILP of equal importance. Building upon this element-centric view, we propose an attention-driven backbone that stacks $T$ layers, each with its own trainable parameters. In each layer $t \in \{1, \dots, T\}$, the model updates the variable embeddings $\mathbf{H}_t^V$ and constraint embeddings $\mathbf{H}_t^C$ via a dual-attention mechanism.

### 3.1. The Dual-Attention Mechanism

The dual-attention mechanism consists of (i) self-attention applied separately to variables and constraints to aggregate same-type information globally, and (ii) cross-attention to model variable–constraint interactions. Stacking these layers yields more expressive embeddings that capture both global context and local dependencies.

**Intra-Type Self-Attention.** We utilize the self-attention mechanism (Vaswani et al., 2017) on the variable set $\mathcal{V}$ and constraint set $\mathcal{C}$ to extract the corresponding embeddings. For the $n$ variables, we first compute the query/key/value matrices as $\mathbf{Q}_V = \mathbf{H}_{t-1}^V W_Q^V$, $\mathbf{K}_V = \mathbf{H}_{t-1}^V W_K^V$, and $\mathbf{V}_V = \mathbf{H}_{t-1}^V W_V^V$, where $W_Q^V, W_K^V, W_V^V$ are trainable parameters, and update the embeddings as:

$$
\begin{aligned}
\mathbf{H}_{\text{self},t}^V &= \text{SelfAttn}_V(\mathbf{Q}_V, \mathbf{K}_V, \mathbf{V}_V) \\
&= \text{softmax}\left(\frac{\mathbf{Q}_V \mathbf{K}_V^\top}{\sqrt{d}}\right) \mathbf{V}_V.
\end{aligned} \quad (3)
$$

The constraint embeddings are computed similarly as:

$$
\begin{aligned}
\mathbf{H}_{\text{self},t}^C &= \text{SelfAttn}_C(\mathbf{Q}_C, \mathbf{K}_C, \mathbf{V}_C) \\
&= \text{softmax}\left(\frac{\mathbf{Q}_C \mathbf{K}_C^\top}{\sqrt{d}}\right) \mathbf{V}_C,
\end{aligned} \quad (4)
$$

where $\mathbf{Q}_C = \mathbf{H}_{t-1}^C W_Q^C$, $\mathbf{K}_C = \mathbf{H}_{t-1}^C W_K^C$, and $\mathbf{V}_C = \mathbf{H}_{t-1}^C W_V^C$. This step allows all variable (or constraint)

nodes to exchange information globally within the same type, regardless of their distance in the bipartite graph.

Self-attention is known to be computationally expensive. Assuming $d \ll n, m$, it incurs $O((n^2 + m^2)d)$ time and $O(n^2 + m^2)$ memory, which forms a major bottleneck for large-scale MILP instances. To address this issue, we employ a simple kernel trick (Deng et al., 2024) to linearize and approximate the standard self-attention as follows:

$$
\widetilde{\mathbf{H}}_{\text{self},t}^V = \frac{\sigma(\mathbf{Q}_V)(\sigma(\mathbf{K}_V^\top)\mathbf{V}_V)}{\sigma(\mathbf{Q}_V)\sum_i \sigma(\mathbf{K}_V^\top(i,:)) + \epsilon} \approx \mathbf{H}_{\text{self},t}^V, \quad (5)
$$

$$
\widetilde{\mathbf{H}}_{\text{self},t}^C = \frac{\sigma(\mathbf{Q}_C)(\sigma(\mathbf{K}_C^\top)\mathbf{V}_C)}{\sigma(\mathbf{Q}_C)\sum_j \sigma(\mathbf{K}_C^\top(j,:)) + \epsilon} \approx \mathbf{H}_{\text{self},t}^C, \quad (6)
$$

where $\sigma$ denotes the sigmoid function to guarantee positive attention scores, and $\epsilon$ is a small constant ($1e - 8$) used to avoid zero denominator. By first computing $\sigma(\mathbf{K}_V^\top)\mathbf{V}_V$ and $\sigma(\mathbf{K}_C^\top)\mathbf{V}_C$ in Eq. (5) and (6), the time complexity is reduced to $O(nd^2 + md^2)$. The precomputation costs $O(d^2)$ memory, the subsequent multiplication with $\sigma(\mathbf{Q})$ requires $O(nd)$ or $O(md)$. Thus, the overall memory complexity is reduced to $O((n + m)d)$.

We use $H$ attention heads with residual connection and layer normalization to get the final embeddings as follows:

$$
\mathbf{H}_{\text{self},t}^V = \text{LN}\left(\mathbf{H}_{t-1}^V + (\overset{H}{\underset{h=1}{\|}} \widetilde{\mathbf{H}}_{\text{self},t,h}^V)\mathbf{W}_t^V\right), \quad (7)
$$

$$
\mathbf{H}_{\text{self},t}^C = \text{LN}\left(\mathbf{H}_{t-1}^C + (\overset{H}{\underset{h=1}{\|}} \widetilde{\mathbf{H}}_{\text{self},t,h}^C)\mathbf{W}_t^C\right). \quad (8)
$$

where $\|$ is concatenation and LN is layer normalization.

**Inter-Type Cross-Attention.** We consider variables and constraints to be of equal importance, and design a bidirectional cross-attention mechanism to analyze the relationship between them. For the variables, we compute the query matrix $\mathbf{Q}_V = \mathbf{H}_{t-1}^V W_Q^V$ using variable embeddings,

and compute the key/value matrices $\mathbf{K}_C = \mathbf{H}_{t-1}^C W_K^C$ and $\mathbf{V}_C = \mathbf{H}_{t-1}^C W_V^C$ using constraint embeddings, where $W_Q^V$, $W_K^C$, and $W_V^C$ are trainable parameters[1]. The variable embeddings are updated as follows:

$$\begin{aligned}
\mathbf{H}_{\mathrm{cross},t}^V &= \mathrm{CrossAttn}_V(\mathbf{Q}_V, \mathbf{K}_C, \mathbf{V}_C) \\
&= \mathrm{softmax}\left(\frac{\mathbf{Q}_V \mathbf{K}_C^\top}{\sqrt{d}}\right) \mathbf{V}_C.
\end{aligned} \quad (9)$$

The constraint embeddings are computed in the same fashion:

$$\begin{aligned}
\mathbf{H}_{\mathrm{cross},t}^C &= \mathrm{CrossAttn}_C(\mathbf{Q}_C, \mathbf{K}_V, \mathbf{V}_V) \\
&= \mathrm{softmax}\left(\frac{\mathbf{Q}_C \mathbf{K}_V^\top}{\sqrt{d}}\right) \mathbf{V}_V,
\end{aligned} \quad (10)$$

where $\mathbf{Q}_C = \mathbf{H}_{t-1}^C W_Q^C$, $\mathbf{K}_V = \mathbf{H}_{t-1}^V W_K^V$, and $\mathbf{V}_V = \mathbf{H}_{t-1}^V W_V^V$.

The above cross-attention has two weaknesses: it ignores the MILP structure explicitly and incurs $O(mnd)$ time and $O(mn)$ memory due to all-to-all interactions, which is prohibitive for large-scale instances. To address this issue, we incorporate the coefficient matrix $\mathbf{A}$ in the MILP problem into the cross-attention computation. As is well known, $\mathbf{A}$ is often sparse, which can be exploited to obtain structural properties and speed up computation. Specifically, we compute attention only on $\mathcal{E}$. For the variable-to-constraint attention, we construct a sparse and coefficient-aware attention score matrix $\mathbf{M}_V$, where $\mathbf{M}_{V,ij} = \exp(p_{ij})$ if $(i,j) \in \mathcal{E}$ and $\mathbf{M}_{V,ij} = 0$ otherwise. Here, $\mathbf{M}_{V,ij}$ denotes the attention score from variable $i$ to constraint $j$ and $p_{ij} = \sum_{k=1}^d [\frac{\mathbf{Q}_{V,i} \odot \mathbf{K}_{C,j}}{\sqrt{d}} \odot (\mathbf{W}_{E,ij} \mathbf{A}_{ij})]_k$, where $\mathbf{W}_{E,ij}$ is trainable parameter. Then we can obtain the sparse variable-to-constraint attention matrix $\mathbf{D}_V^{-1} \mathbf{M}_V$, where $\mathbf{D}_V \in \mathbb{R}^{n \times n}$ is the normalized diagonal matrix on the variable side with $\mathbf{D}_{V,ii} = \sum_{j=1}^m \mathbf{M}_{V,ij} + \epsilon$, and $\epsilon = 1e{-}8$ is a small constant. The resulting update is given by:

$$\mathbf{H}_{\mathrm{cross},t}^V = \mathbf{D}_V^{-1} \mathbf{M}_V \mathbf{V}_C. \quad (11)$$

Similarly, the constraint embedding update in the constraint-to-variable attention part is as follows:

$$\mathbf{H}_{\mathrm{cross},t}^C = \mathbf{D}_C^{-1} \mathbf{M}_C \mathbf{V}_V. \quad (12)$$

By introducing the sparse matrix $\mathbf{A}$ with $e$ nonzeros, the time complexity and memory complexity of our cross-attention part are reduced to $O(ed)$ and $O(e)$ respectively.

As with self-attention, we use $H$ attention heads and apply residual connections and layer normalization to obtain the

final embeddings:

$$\mathbf{H}_{\mathrm{cross},t}^V = \mathrm{LN}\left(\mathbf{H}_{t-1}^V + (\overset{H}{\underset{h=1}{\|}} \mathbf{H}_{\mathrm{cross},t,h}^V) \mathbf{W}_t^V\right), \quad (13)$$

$$\mathbf{H}_{\mathrm{cross},t}^C = \mathrm{LN}\left(\mathbf{H}_{t-1}^C + (\overset{H}{\underset{h=1}{\|}} \mathbf{H}_{\mathrm{cross},t,h}^C) \mathbf{W}_t^C\right). \quad (14)$$

**Parallel Inference and Feature Fusion.** A key design in our dual-attention is the parallel inference scheme, where the self-attention and cross-attention modules work concurrently. Conventional BGNN uses two stage message passing (variables→constraints, constraints→variables) to extract node embeddings. The constraint nodes practically work as intermediate messengers, forming information bottlenecks and incurring over-squashing. In contrast, our parallel design ensures that the intra-type and inter-type relationships are learned *separately* before fused together, effectively reducing interference between the two aspects and mitigating the information bottleneck issue. As the final operation in our dual-attention mechanism, we fuse the learned embeddings as follows:

$$\mathbf{H}_t^V = \mathrm{FFN}\big(\mathrm{MLP}^V(\mathbf{H}_{\mathrm{self},t}^V \| \mathbf{H}_{\mathrm{cross},t}^V)\big), \quad (15)$$

$$\mathbf{H}_t^C = \mathrm{FFN}\big(\mathrm{MLP}^C(\mathbf{H}_{\mathrm{self},t}^C \| \mathbf{H}_{\mathrm{cross},t}^C)\big), \quad (16)$$

where FFN denotes Position-wise Feed-Forward Network.

**Overall Complexity.** Given an MILP instance with $n$ variables, $m$ constraints, and $e$ nonzeros in $\mathbf{A}$, the time and memory complexities of a dual-attention layer are $O((m+n+e)d^2)$ and $O((m+n+e)d)$, respectively, supporting scalability to large instances. The inference time of our method is provided in Appendix A.4.

### 3.2. Application to Downstream Tasks

After $T$ dual-attention layers, we obtain variable and constraint embeddings $\mathbf{H}_T^V$ and $\mathbf{H}_T^C$ for downstream tasks. We consider the following representative learning tasks.

**Instance-Level Prediction.** These tasks focus on predicting a key property of an MILP instance, often in the form of supervised learning. We apply mean pooling to the $n$ variable embeddings and $m$ constraint embeddings respectively to obtain two vectors, which are then concatenated and fed into an MLP to produce the final prediction:

$$\Phi_{\mathrm{inst}} = \mathrm{MLP}\left(\mathrm{MEAN}(\mathbf{H}_T^V) \| \mathrm{MEAN}(\mathbf{H}_T^C)\right). \quad (17)$$

where $\mathrm{MEAN}(\mathbf{H}_T^V) = \frac{1}{n} \sum_{i=1}^n \mathbf{h}_{T,i}^V$, $\mathrm{MEAN}(\mathbf{H}_T^C) = \frac{1}{m} \sum_{j=1}^m \mathbf{h}_{T,j}^C$. Here we consider two representative tasks following (Chen et al., 2023): 1) instance feasibility prediction, which is a binary classification task and $\Phi_{\mathrm{inst}} \in [0,1]$ is the probability of being feasible; 2) optimal objective value prediction, which is a regression task and $\Phi_{\mathrm{inst}} \in \mathbb{R}$ is the predicted objective value.

---

[1]The trainable parameters of the cross-attention module differ from those of the self-attention module. For notational simplicity, we omit the explicit dependence on the module.

**Element-Level Prediction.** Another type of task is predicting element-level targets directly from the learned variable or constraint embeddings. The most commonly studied task in this direction is to predict the values of binary variables in the optimal solution $\mathbf{x}^*$ via supervised learning:

$$\Phi_{\text{sol}} = \text{MLP}\left(\mathbf{H}_T^{BV}\right), \tag{18}$$

where $\mathbf{H}_T^{BV} = \left[\mathbf{h}_{T,i}^V \big| \text{if } \mathbf{x}_i \text{ is binary}\right]$ is the collection of binary variable embeddings, and $\Phi_{\text{sol}} \in [0,1]^{n_b}$ is the probability vector of all the $n_b$ binary variables. Binary variables often make up the majority in practice (Ding et al., 2020; Scavuzzo et al., 2024), so accurate predictions can substantially enhance the solver's efficiency. A representative method is Prediction-and-Search (PaS) (Han et al., 2023). It fixes the $k_0$ variables with the lowest predicted probabilities to 0 and the $k_1$ with the highest to 1, while allowing up to $\Delta$ flips via an additional linear constraint. Alternating Prediction Correction Neural Solving Framework (Apollo) (Liu et al., 2025) extends PaS by using a short solver probing timeframe to correct uncertain or inaccurate predictions, further improving acceleration.

**Solving State-Level Prediction.** Branch-and-bound (B&B) algorithms are the core of modern MILP solvers. It is a complicated tree search process involving many decisions that can affect the solving efficiency. Deep learning can be of great value here by predicting high-quality decisions at each B&B solving state, which can naturally be formulated as a sequential decision-making problem. We apply our method to learning-to-branch (Gasse et al., 2019), the most widely studied task of this type, which aims to predict the branching variable at each B&B node to minimize the search tree size. This can be achieved by associating solving state-dependent features for variables and constraints, employing our neural architecture to extract corresponding embeddings, and then conducting the following computation:

$$\Phi_{\text{state}} = \text{softmax}\left(\text{MLP}(\mathbf{H}_{\text{state},T}^V), \mathbf{P}_{\text{state}}\right) \tag{19}$$

where $\mathbf{H}_{\text{state},T}^V$ denotes the state-dependent variable embeddings, and $\mathbf{P}_{\text{state}}$ is a $n$-dimensional binary vector to mask out variables that already have fixed values at that state, such that the resulting $\Phi_{\text{state}}$ represents a distribution over candidate variables. This policy network can be trained via imitation or reinforcement learning.

# 4. Experiments

This section compares our attention-based model with BGNN on the three tasks in Section 3.2. We use the same architecture with $T = 4$ dual-attention layers, $H = 2$ attention heads and $d = 64$ embedding dimension throughout the experiments. For BGNN, we follow the configuration in all our baselines (Chen et al., 2023; Han et al., 2023; Liu et al., 2025; Gasse et al., 2019) with 2 layers

and $d = 64$. Other implementation details are provided in Appendix E. Our source code is available at `https://github.com/hpx2024/Dual-Attention`.

## 4.1. Performance on Instance-Level Prediction

In this part, we follow (Chen et al., 2023) to assess feasibility and optimal objective value prediction. As noted in (Chen et al., 2023), the foldable MILPs are challenging for vanilla BGNN; thus, an additional random feature is appended to each variable and constraint vector to enhance the power of BGNN, which is also added to our model for fair comparison. Using the procedure in (Chen et al., 2023) (see Appendix C.1), we generate MILP instances with $n = 20$ variables and $m = 6$ constraints, and split them into two datasets $\mathcal{D}_1$ (unfoldable) and $\mathcal{D}_2$ (foldable). Both sets have 2000 instances, among which we use 1000 instances for training and the remaining for testing. Labels are obtained with the SCIP solver (Bolusani et al., 2024) by determining the feasibility and solving feasible instances to optimality.

Table 1. Feasibility and optimal objective value prediction.

| Dataset | Method | Feas. E-Rate↓ | Obj. MSE↓ |
|---|---|---|---|
| $\mathcal{D}_1$ (Unfoldable) | BGNN | **0.0%** | 3.975e-09 |
| | Ours | **0.0%** | **3.889e-14** |
| $\mathcal{D}_2$ (Foldable) | BGNN | 6.0% | 1.178e-11 |
| | Ours | **0.0%** | **2.312e-13** |

Results in Table 1 show that for feasibility prediction, both models can achieve perfect accuracy on the simpler dataset $\mathcal{D}_1$. However, on the more challenging dataset $\mathcal{D}_2$, our model can still correctly predict the feasibility of all testing instances, whereas the model in (Chen et al., 2023) incurs a 6% error rate. For optimal objective value prediction, the Mean Squared Error (MSE) of our method is five and two orders of magnitude smaller than that of the enhanced BGNN in (Chen et al., 2023) on $\mathcal{D}_1$ and $\mathcal{D}_2$, respectively. These results demonstrate the superior accuracy and robustness of our model in instance-level prediction tasks.

Table 2. Performance evaluation on binary variable prediction.

| Dataset | Method | MCC↑ | M-F1↑ | MSE↓ | E-Rate↓ |
|---|---|---|---|---|---|
| IP | BGNN | 0.0156 | 0.5060 | **0.0900** | 20.88% |
| | Ours | **0.0178** | **0.5089** | **0.0900** | **17.72%** |
| WA | BGNN | 0.7972 | 0.8986 | 0.0492 | 6.83% |
| | Ours | **0.8411** | **0.9205** | **0.0386** | **5.32%** |
| MIS | BGNN | 0.6431 | 0.8215 | 0.1234 | 17.72% |
| | Ours | **0.6920** | **0.8460** | **0.1058** | **15.29%** |
| CA | BGNN | 0.3488 | 0.6742 | 0.1429 | 23.10% |
| | Ours | **0.3825** | **0.6911** | **0.1383** | **21.86%** |

*Table 3.* Average PG and PI (for each problem, the value within parenthesis is the BKV).

| Method | IP (12.24) | | WA (700.96) | | MIS (1371.74 ) | | CA (220061.61) | | Avg. Impr. over IP/WA/MIS/CA | |
|---|---|---|---|---|---|---|---|---|---|---|
| | PG↓ | PI↓ | PG↓ | PI↓ | PG↓ | PI↓ | PG↓ | PI↓ | PG Avg Impr.(%) | PI Avg Impr.(%) |
| Gurobi | 6.944% | 144.45 | 0.026% | 2.74 | **0%** | 0.46 | **0.222%** | **4.34** | - | - |
| PaS-BGNN | 7.435% | 146.27 | 0.021% | 2.64 | **0%** | 0.02 | 0.246% | 4.73 | - | - |
| PaS-Ours | 5.556% | 106.13 | **0.016%** | **2.60** | **0%** | **0.01** | 0.232% | 4.29 | 14.462% | 22.065% |
| Apollo-BGNN | 4.984% | 112.36 | 0.020% | 2.75 | 0.008% | 0.53 | 0.316% | 5.21 | - | - |
| Apollo-Ours | **4.167%** | **102.87** | 0.019% | 2.73 | **0%** | 0.44 | 0.249% | 4.53 | 36.143% | 9.802% |

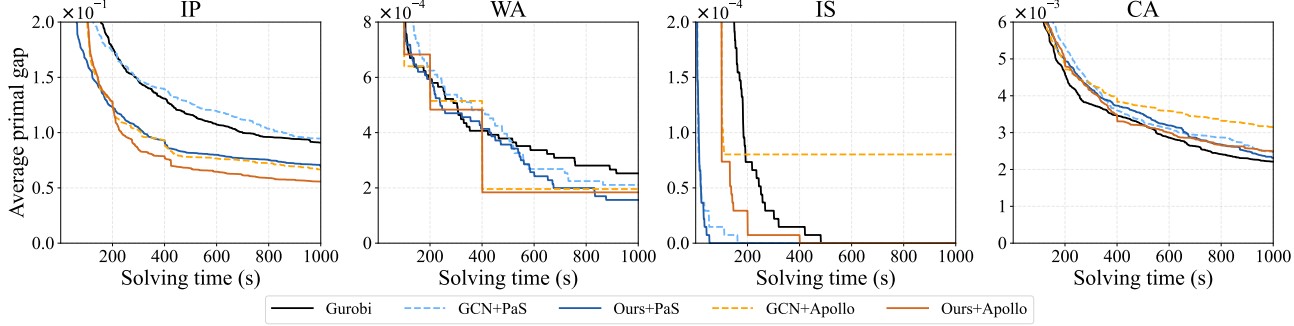

*Figure 2.* **Average Primal Gap (PG) of each method as a function of solving time.**

## 4.2. Performance on Element-Level Prediction

In this part, we study binary variable prediction under the setting of PaS (Han et al., 2023) and Apollo (Liu et al., 2025). We use four benchmarks: balanced Item Placement (IP), Workload Appointment (WA), Maximum Independent Set (MIS), and Combinatorial Auction (CA). The first two are from the NeurIPS ML4CO 2021 competition (Gasse et al., 2022), while the latter two are generated using the Ecole library (Prouvost et al., 2020). For each problem, we use 400 instances for training and 100 for testing. Detailed problem introduction and data generation method can be found in Appendix C.2. We use Gurobi (LLC Gurobi Optimization, 2021) with a 3,600-second time limit to collect supervised training labels and as the solver in the following predict-and-search experiments. BGNN (also utilized in Apollo (Liu et al., 2025)) and our model are trained within the PaS framework (Han et al., 2023).

We first evaluate binary variable prediction accuracy. This is a binary classification task with no intrinsic positive or negative class and potential class imbalance, with the 0/1 label ratios of 9:1 (IP), 1:4 (WA), 3:1 (MIS), and 1:1 (CA). Therefore, we use macro F1-score (M-F1), Matthews correlation coefficient (MCC), the MSE between $\Phi_{sol}$ and the optimal solution $x^*$, and the rate of incorrectly predicted variables (E-Rate) as metrics. As shown in Table 2, our attention-based model consistently outperforms BGNN across all datasets and all metrics, correctly predicts 15.1%, 22.2%, 13.7% and 5.4% more variables on IP, WA, MIS and CA respectively, demonstrating a much stronger prediction ac-

curacy. We further visualize the probability distribution of binary variable predictions on MIS in Appendix A.1.

Next, we evaluate how solution prediction affects downstream MILP solving. The trained BGNN and our model are integrated into the PaS (Han et al., 2023) and Apollo (Liu et al., 2025) frameworks to solve 100 test instances with Gurobi under a 1000-second limit. The original Gurobi with a 3600-second time limit serves as a reference for selecting the best-known value (BKV) from all methods. We report the geometric mean of the Primal Gap (PG) and the Primal Integral (PI) (Berthold, 2013) as evaluation metrics. PG measures final solution quality relative to the BKV and PI quantifies the solver's efficiency in converging toward the optimum over time. Further details are provided in Appendix D. Table 3 and Fig. 2 show that our attention-based architecture consistently outperforms the two BGNN-based counterparts across all problems, improving both final solution quality and convergence speed.

## 4.3. Performance on Solving State-Level Prediction

In this part, we evaluate our attention-based architecture within the Imitation Learning (IL) framework of (Gasse et al., 2019). It learns a fast neural approximation of the Strong Branching (SB) strategy, which produces compact search trees but is computationally expensive. While many subsequent works study learning-to-branch, they mostly focus on training mechanisms and still rely on the BGNN backbone from (Gasse et al., 2019). We adopt this standard setup for a controlled comparison across architectures.

*Table 4.* Imitation learning accuracy on the test sets (higher is better).

| Method | SC | | | CA | | | CFL | | | MIS | | |
|---|---|---|---|---|---|---|---|---|---|---|---|---|
| | acc@1 | acc@5 | acc@10 | acc@1 | acc@5 | acc@10 | acc@1 | acc@5 | acc@10 | acc@1 | acc@5 | acc@10 |
| BGNN | 70.35 | 93.16 | 98.39 | 70.07 | 93.11 | 98.49 | 53.50 | 88.70 | 96.40 | 81.00 | 92.40 | 95.70 |
| Ours | **71.51** | **93.75** | **98.61** | **70.20** | **93.90** | **98.60** | **57.20** | **90.30** | **96.90** | **82.70** | **93.20** | **96.10** |

Following (Gasse et al., 2019), we evaluate on four NP-hard benchmarks: Set Covering (SC), Combinatorial Auctions (CA), Capacitated Facility Location (CFL), and Maximum Independent Set (MIS), with formulations and instance generation in Appendix C. For each problem, we generate 10,000 instances for training and 40 for testing. Additionally, we generate 40 instances larger and more challenging than the training set instances to form the transfer set. Since learning-to-branch requires direct access to the solver's internal logic, we use the open-source solver SCIP (Bolusani et al., 2024) as the environment, and collect SB demonstrations using its built-in implementation.

We compare with BGNN in the learning-to-branch task from two aspects as in (Gasse et al., 2019). First, in terms of prediction accuracy, Table 4 reports the acc@1, acc@5, and acc@10 metrics, representing the proportion of times that the correct SB label appears in the top-1, top-5, and top-10 ranked predictions. Under the same training framework, our model consistently achieves higher accuracy across all problems. Second, the learned policy is embedded in SCIP to solve the test and transfer instances. For each instance, we evaluate each policy under five random seeds. As shown in Table 5, our method yields smaller B&B trees than BGNN, with larger gains on the transfer set.

*Table 5.* Comparison of B&B tree size (i.e., number of nodes).

| Method | Test | | | |
|---|---|---|---|---|
| | SC | CA | CFL | MIS |
| BGNN | 65.4±11% | 57.8±16% | 350.4±43% | 48.6±47% |
| Ours | **64.9±10%** | **57.3±13%** | **323.1±39%** | **48.4±45%** |

| Method | Transfer | | | |
|---|---|---|---|---|
| | SC | CA | CFL | MIS |
| BGNN | 160.9±10% | 836.5±14% | 545.7±48% | 10280.5±73% |
| Ours | **152.3±7%** | **811.7±12%** | **512.5±48%** | **8938.5±42%** |

### 4.4. Mechanism Analysis

To understand why our attention-based architecture outperforms BGNN, we use the binary variable prediction task to conduct a fine-grained analysis in deep representation learning and long-range dependency modeling.

#### 4.4.1. EXTRACTING DEEP REPRESENTATIONS

We analyze BGNN and our model on MIS under different numbers of layers (from 1 to 5) in Fig. 3. First, Fig. 3a and 3e show violin plots of Z-score normalized embedding values obtained at layers 1 and 5. For BGNN, embedding values at the 5th layer are more concentrated than the 1st layer, indicating a trend of collapsing into similar representations. In contrast, our model exhibits minimal variation in embedding value distributions across shallow and deep layers, indicating a much stronger ability in preserving feature diversity and mitigating the over-smoothing issue.

Second, Fig. 3b and 3f show the t-SNE visualization of the embeddings obtained at the 1st and 5th layer projected to the same 2D space. For BGNN, embeddings tend to collapse into one-dimensional manifolds, with a clear decrease in the spatial coverage from shallow to deep layers. This indicates a decline of diversity in the feature space and further validates the phenomenon of over-smoothing. In contrast, our model exhibits a much more diverse distribution in the feature space, effectively mitigating the feature collapse issue of BGNN.

Third, Fig. 3c and 3g report the gradient flow analysis by plotting the $\ell_2$-norm of gradient at each layer, normalized by the first layer. A clear distinction between the two models can be observed. BGNN's gradient rapidly and almost monotonically decays from layer 1 to 5, which severely affects backpropagation through deeper layers and largely explains why it struggles to utilize more than 2 layers. In contrast, the gradient strength in our model stably increases and peaks at the 4th layer before a moderate decline at the 5th layer, showing that our design enables effective backpropagation through more layers, enabling the network to learn deeper and richer representations.

Finally, we evaluate the prediction performance of each method under different depths. Specifically, we train each model five times per layer setting and plot the mean Macro-F1 with standard deviation in Fig. 3d and 3h. Evidently, BGNN becomes unstable beyond two layers and the performance degrades sharply at five layers. Hence, following most prior works, we use a two-layer BGNN in all experiments. In contrast, our model remains stable and improves with depth. We uniformly use four layers as a trade-off between representation learning power and model complexity.

#### 4.4.2. CAPTURING LONG-RANGE DEPENDENCIES

We perform gradient-based attribution (Ancona et al., 2018) to analyze the capability of the two-layer BGNN and our four-layer attention architecture in capturing long-range de-

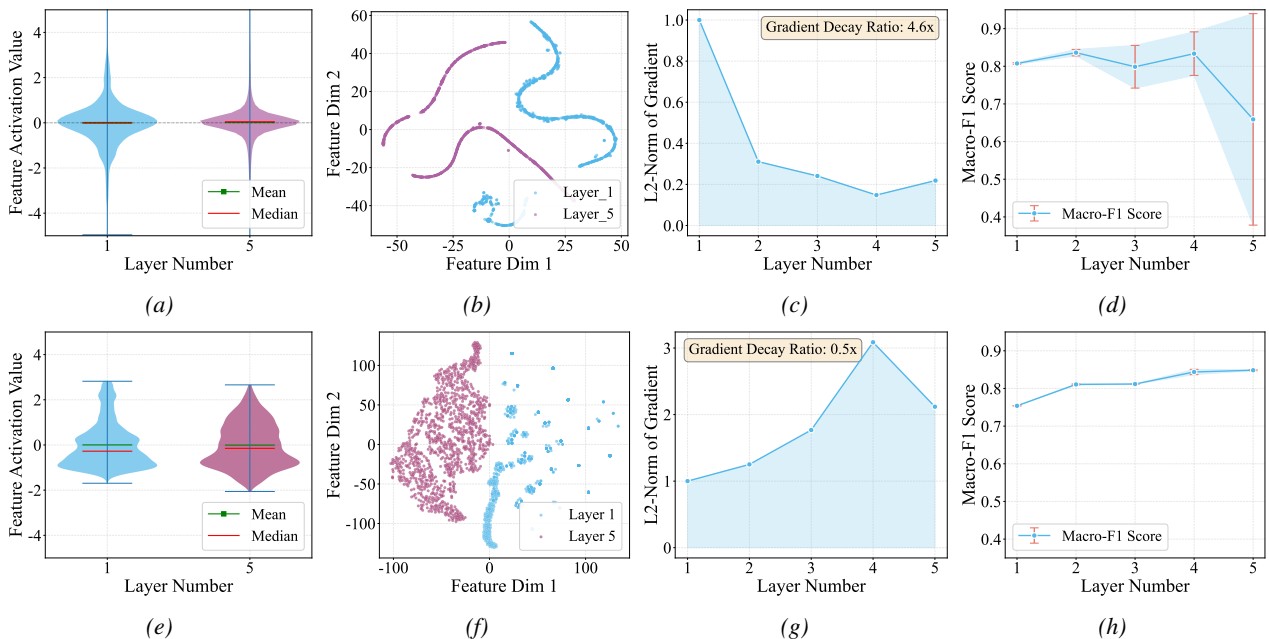

*Figure 3.* **Comparison of the performance in extracting deep representations.** (a) and (e) depict the normalized embedding values of the BGNN and our model at the 1st and 5th layer. (b) and (f) depict the t-SNE visualization of embeddings at the 1st and 5th layer of BGNN and our model. (c) and (g) depict the decay of gradients in the BGNN and our model as the network deepens. (d) and (h) depict the mean Macro-F1 score with standard deviation of training BGNN and our model five times under different numbers of layers.

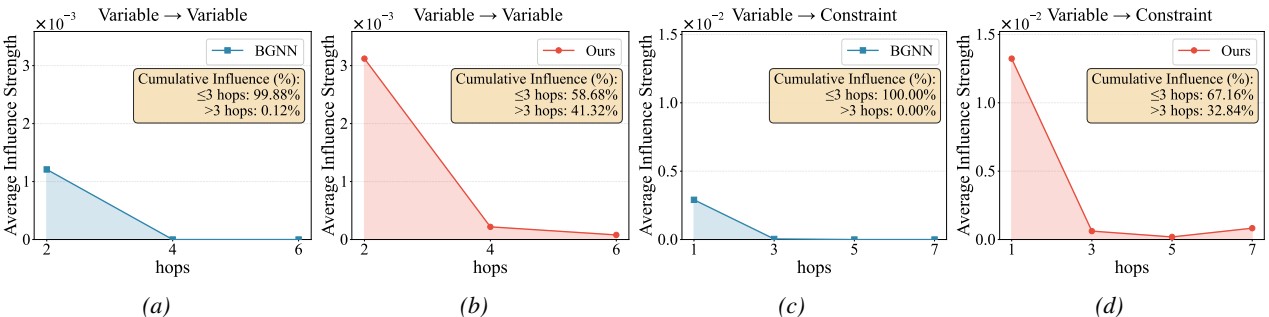

*Figure 4.* **Analysis of long-range dependency modeling in BGNN and our attention-based architecture.** Gradient-based attribution is used to quantify the influence strength between a target variable and other nodes at different distances. Panels (a) and (b) show average variable-to-variable influence for BGNN and our model, while panels (c) and (d) show average variable-to-constraint influence.

pendencies. We randomly draw an instance from the WA test set, and the longest distance between two nodes in the underlying bipartite graph is 7 hops. We backpropagate each variable prediction $x_i$ and compute the $\ell_2$-norm of feature gradients on other variable or constraint nodes as influence scores. Fig. 4a and 4b illustrate the average influence of variables (i.e., nodes that are 2, 4, 6 hops away), whilst Fig. 4c and 4d demonstrate the average influence of constraints (nodes that are 1, 3, 5, 7 hops away) of the respective models. It is evident that for BGNN, the influence of nodes decays sharply with distance, and the variables or constraints beyond 3 hops exert negligible impact on BGNN's prediction since the influence almost vanishes. Though the two-layer BGNN theoretically propagates information across up to 4 hops, in practice the signal from 4-hop nodes is almost com-

pletely suppressed. In contrast, our attention architecture maintains substantial non-zero influence even at the farthest 7 hops, demonstrating its ability to effectively capture long-range dependencies that BGNN fails to exploit.

### 4.5. Comparison with Advanced Architectures

We further examine the performance of our attention-driven architecture by comparing with stronger transformer-style and advanced graph backbones, including OPTFM (Yuan et al., 2025), MAGCN (Toan & Van-Hop, 2025), GCON (Wenkel et al., 2025), Polynormer (Deng et al., 2024), and GNN-SSM (Arroyo et al., 2025). OPTFM and MAGCN are concurrent works developed for MILP representation learning and their architectures are directly comparable.

GCON is a sophisticated GNN developed for COPs defined on graphs. Polynormer is an advanced graph transformer. GNN-SSM is a state-of-the-art method in overcoming the over-smoothing and over-squashing issues of GNN. Since Polynormer, GCON, and GNN-SSM are originally developed for homogeneous graphs, we adapt them in a bipartite $v \rightarrow c \rightarrow v$ fashion to suit MILP's bipartite structure. These models are evaluated on the four binary variable prediction datasets in Section 4.2 and further integrated into the prediction-and-search framework for downstream solving. For fairness, we tune the depth of each of these strong backbones on the WA dataset to their optimal performance (see Appendix E.4 for details).

*Table 6.* Extended comparison on binary variable prediction across IP, WA, MIS, and CA.

| Dataset | Method | MCC↑ | M-F1↑ | MSE↓ | E-Rate↓ |
|---|---|---|---|---|---|
| IP | BGNN | 0.0156 | 0.5060 | 0.0900 | 20.88% |
| | OPTFM | 0.0000 | 0.4737 | **0.0899** | **10.00%** |
| | Polynormer | 0.0135 | 0.5066 | 0.0901 | 17.07% |
| | MAGCN | 0.0148 | 0.5046 | 0.0899 | 14.66% |
| | GCON | 0.0078 | 0.4785 | 0.0899 | 10.32% |
| | GNN-SSM | **0.0186** | 0.5065 | 0.0899 | 14.61% |
| | Ours | 0.0178 | **0.5089** | 0.0900 | 17.72% |
| WA | BGNN | 0.7972 | 0.8986 | 0.0492 | 6.83% |
| | OPTFM | 0.7155 | 0.8576 | 0.0724 | 9.78% |
| | Polynormer | 0.7408 | 0.8704 | 0.0649 | 8.75% |
| | MAGCN | 0.7971 | 0.8985 | 0.0498 | 6.89% |
| | GCON | 0.7991 | 0.8995 | 0.0493 | 6.72% |
| | GNN-SSM | 0.8022 | 0.9011 | 0.0480 | 6.65% |
| | Ours | **0.8411** | **0.9205** | **0.0386** | **5.32%** |
| MIS | BGNN | 0.6431 | 0.8215 | 0.1234 | 17.72% |
| | OPTFM | 0.5169 | 0.7510 | 0.1680 | 24.85% |
| | Polynormer | 0.6008 | 0.8004 | 0.1388 | 19.79% |
| | MAGCN | 0.6192 | 0.8096 | 0.1293 | 18.90% |
| | GCON | 0.6868 | 0.8434 | 0.1094 | 15.56% |
| | GNN-SSM | 0.6892 | 0.8451 | 0.1073 | 15.42% |
| | Ours | **0.6920** | **0.8460** | **0.1058** | **15.29%** |
| CA | BGNN | 0.3488 | 0.6742 | 0.1429 | 23.10% |
| | OPTFM | 0.2126 | 0.6058 | 0.1620 | 28.44% |
| | Polynormer | 0.2402 | 0.6182 | 0.1603 | 28.51% |
| | MAGCN | 0.3415 | 0.6703 | 0.1440 | 23.64% |
| | GCON | 0.3248 | 0.6616 | 0.1469 | 24.49% |
| | GNN-SSM | 0.3643 | 0.6839 | 0.1400 | 22.65% |
| | Ours | **0.3825** | **0.6911** | **0.1383** | **21.86%** |

As shown in Table 6, our method achieves the strongest overall prediction performance. In particular, it obtains the best results on WA, MIS, and CA across all four metrics, and remains competitive on IP, where the extremely imbalanced 0/1 label ratio (9:1) makes E-Rate less informative. This consistent advantage over stronger competing backbones suggests that the dual-attention design provides more effective MILP representations than directly adapting existing transformer-style or advanced GNN architectures for MILP representation learning.

We further report the downstream solving performance in Tables 7 and 8. It can be observed that our method obtains the best average improvement in both PG and PI. Specifi-

cally, it achieves an average PG improvement of 9.047% and an average PI improvement of 20.568% over BGNN under the prediction-and-search framework. These results indicate that the improved prediction quality can be effectively translated into better downstream solving behavior, further supporting the practical value of the proposed backbone.

*Table 7.* Average PG under the prediction-and-search framework.

| Method | IP | WA | MIS | CA | Impr. |
|---|---|---|---|---|---|
| BGNN | 11.958% | 3.867% | **0.000%** | 0.268% | – |
| OPTFM | **9.380%** | 3.881% | 0.023% | 0.237% | -16.827% |
| Polynormer | 10.033% | 3.892% | **0.000%** | 0.272% | 3.523% |
| MAGCN | 11.444% | 3.878% | **0.000%** | 0.290% | -0.985% |
| GCON | 11.217% | **3.796%** | **0.000%** | 0.280% | 0.906% |
| GNN-SSM | 9.837% | 3.911% | **0.000%** | 0.300% | 1.156% |
| Ours | 9.409% | 3.861% | **0.000%** | **0.229%** | **9.047%** |

*Table 8.* Average PI under the prediction-and-search framework.

| Method | IP | WA | MIS | CA | Impr. |
|---|---|---|---|---|---|
| BGNN | 173.32 | 41.18 | 0.02 | 5.18 | – |
| OPTFM | 134.47 | **40.32** | 0.24 | 4.75 | -16.816% |
| Polynormer | 141.65 | 45.94 | 0.18 | 5.20 | -23.432% |
| MAGCN | 151.95 | 43.44 | 0.02 | 5.82 | -1.418% |
| GCON | 157.16 | 43.62 | 0.06 | 5.83 | -27.286% |
| GNN-SSM | 147.58 | 44.43 | 0.10 | 5.96 | -27.041% |
| Ours | **132.20** | 41.12 | **0.01** | **4.74** | **20.568%** |

### 4.6. Additional Results

Due to the space constraint, the ablation experiments of our method and comparisons with other alternative designs are presented in Appendix A.2. We also provide an experiment on the well-known MIPLIB datasets in Appendix A.3.

## 5. Conclusion

In this study, we present an attention-driven representation learning method for MILP built on an element-centric view of variables and constraints. Our dual-channel design performs intra-type self-attention and symmetric inter-type cross-attention in parallel, enabling global information exchange and richer variable–constraint interaction modeling beyond the locality of GNN message passing. Across three representative task settings, the proposed model consistently improves prediction quality and demonstrates strong out-of-distribution generalization. More broadly, our results suggest attention-based mechanisms as a promising foundation for learning-assisted combinatorial optimization beyond graph-centric paradigms.

## Accessibility

Our implementation code and detailed reproduction instructions are available at https://github.com/hpx2024/Dual-Attention.

## Acknowledgments

This work is supported by the National Natural Science Foundation of China under Grant 62473233, and the Guangdong Basic and Applied Basic Research Foundation under Grant 2025A1515011704.

## Impact Statement

This paper presents work whose goal is to advance machine learning for combinatorial optimization, in particular learning-based representations for mixed-integer linear programming. There are many potential societal consequences of our method, none of which we feel must be specifically highlighted here.

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

# A. Additional Results

## A.1. Predicted Probability Analysis

Fig. 5 compares the distribution of predicted probabilities for BGNN and our attention-driven architecture on the MIS problem. For BGNN, the predicted probabilities for true-0 and true-1 variables exhibit substantial overlap, with many samples falling in the ambiguous 0.3–0.7 range, indicating low confidence and limited discriminative ability. In contrast, our model produces highly concentrated and well-separated distributions: true-0 variables are sharply peaked near 0, and true-1 variables near 1, with significantly smaller overlap in the middle range. This demonstrates that the attention-based architecture yields more calibrated and confident predictions, and is significantly more effective at capturing the structural patterns that determine binary variable values in MILPs.

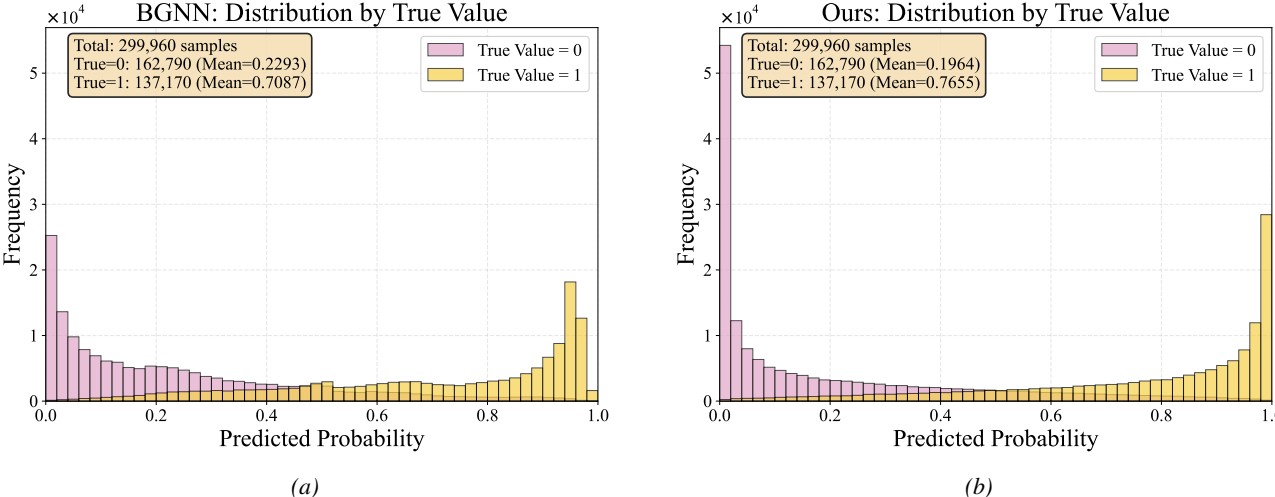

*(a)*                                                     *(b)*

*Figure 5.* **Distribution of predicted probabilities for binary variables.** Histograms show the predicted probability of each binary variable being 1, separated by true labels, for BGNN (a) and our attention-based architecture (b). BGNN exhibits more overlap between the two distributions, with many samples lying in the ambiguous 0.3–0.7 region. In contrast, our model produces sharply separated and highly concentrated distributions near 0 and 1, indicating a stronger discriminative power.

## A.2. Ablation Studies

In this section, we perform ablation studies on the representative binary variable prediction task. We first perform architectural ablation experiments on the WA dataset by independently removing the self-attention and cross-attention modules from our architecture. Table 9 shows that removing either component leads to a significant performance drop, demonstrating that both modules contribute critically and that their joint use is essential for achieving the full effectiveness of our model.

*Table 9.* Ablation results on binary variable prediction using the WA dataset

| Method | MCC↑ | M-F1↑ | MSE↓ | E-Rate↓ |
|---|---|---|---|---|
| Ours | **0.8411** | **0.9205** | **0.0386** | **5.32%** |
| Ours w/o self | 0.7985 | 0.8992 | 0.0486 | 6.76% |
| Ours w/o cross | 0.5724 | 0.7860 | 0.0984 | 14.18% |

Next, we perform alternative design ablation on WA to evaluate other ways of alleviating the limitation of BGNN. First, we employ Graph Attention Network (GATv2) (Brody et al., 2022) to enhance the learning power of BGNN by allowing each node to assign adaptive weights to its neighboring nodes through attention, denoted as BGAT. Second, we augment BGNN by adding an intra-type self-attention module to each of its two message-passing steps in Eq. (2), denoted as BGNN+self. This design allows variables and constraints to globally interact with elements of the same type while retaining the original sequential bipartite propagation scheme. As shown in Table 10, these alternative designs bring only limited improvements over BGNN. Although BGAT provides more flexible local aggregation and BGNN+self introduces global same-type interactions, both variants still follow the graph-based bipartite update scheme and rely on sequential message aggregation. In contrast, our method moves away from the conventional graph-centric propagation view and models intra-type and inter-type interactions in parallel, enabling more effective representation learning for MILP.

*Table 10.* Alternative design ablation on the WA Dataset

| Method | MCC↑ | M-F1↑ | MSE↓ | E-Rate↓ |
|---|---|---|---|---|
| BGNN | 0.7972 | 0.8986 | 0.0492 | 6.83% |
| BGAT | 0.7947 | 0.8973 | 0.0492 | 6.89% |
| BGNN+self | 0.8035 | 0.9017 | 0.0488 | 6.60% |
| Ours | **0.8411** | **0.9205** | **0.0386** | **5.32%** |

## A.3. MIPLIB Experiments

To further demonstrate the practicality of our attention-based model, we conducted binary variable prediction experiments on MIPLIB (Gleixner et al., 2021), a well-recognized benchmark with 240 challenging instances from various real-world scenarios. Binary variables are the majority in MIPLIB: 164 out of the 240 instances are pure binary, and in 44 out of the remaining 76 instances, more than 90% of the integer variables are binary (Ding et al., 2020). This underscores the practical relevance of accurately predicting binary variables in MILP.

We use 215 MIPLIB instances for evaluation, with 25 instances excluded due to three reasons: 1) no binary variables after preprocessing, 2) no feasible solution found after 3,600 seconds Gurobi solving, and 3) out-of-memory. The specific excluded instances are listed in Table 11. All the remaining instances are used in our experiments, among which the largest problem contains 550,539 variables, 1,484 constraints, and 1,101,078 nonzeros.

*Table 11.* Excluded MIPLIB instances

| Issue | Instance name |
|---|---|
| No binary variable after preprocessing (12 instances) | buildingenergy, ex9, ex10, enlight_hard, gen-ip002, gen-ip054, n5-3, neos-3024952-loue, neos-3083819-nubu, ns1952667, supportcase12, supportcase42 |
| No feasible solution after 3,600 seconds Gurobi solving (10 instances) | supportcase19, bnatt500, cryptanalysiskb128n5obj14, cryptanalysiskb128n5obj16, fhnw-binpack4-4, neos-2075418-temuka, neos-3988577-wolgan, neos859080, rail02, supportcase22 |
| Out-of-memory (3 instances) | square47, neos-3402454-bohle, neos-5114902-kasavu |

Given the substantial heterogeneity of MIPLIB instances, directly training prediction models is difficult. We therefore evaluate the out-of-distribution generalization ability of models pre-trained on the MIS, CA, and WA datasets. Models trained on IP are excluded because the positional encoding used to break symmetries in IP is not generally applicable to MIPLIB instances. It is worth noting that the ratio of 0 to 1 labels in the MIPLIB benchmark is approximately 42:1, which is extremely imbalanced. Results in Table 12 demonstrate that our attention-based model significantly outperforms BGNN, regardless of the pre-training sources. The lower error of BGNN-CA over Ours-CA is misleading: under the extreme class imbalance, predicting all variables as 0 yields superficially low error rates but provides no meaningful discriminative ability, as reflected by its near-zero MCC (0.0001) and low macro-F1 score (0.4987). In contrast, our CA model maintains significantly better predictive power despite the dataset shift and label imbalance. Overall, these results demonstrate that our attention-based architecture generalizes robustly to MIPLIB instances that differ significantly from the training distributions, highlighting its potential for practical deployment in real-world MILP solving.

*Table 12.* Performance of binary variable prediction on the MIPLIB benchmark

| Method | MCC↑ | M-F1↑ | MSE↓ | E-Rate↓ |
|---|---|---|---|---|
| BGNN-MIS | 0.0846 | 0.5276 | 0.1997 | 12.27% |
| Ours-MIS | **0.1490** | **0.5730** | **0.0760** | **7.35%** |
| BGNN-CA | 0.0001 | 0.4987 | 0.1615 | **8.03%** |
| Ours-CA | **0.1508** | **0.5728** | **0.1293** | 10.29% |
| BGNN-WA | 0.0099 | 0.5031 | **0.2412** | 12.36% |
| Ours-WA | **0.1053** | **0.5526** | 0.5174 | **9.22%** |

## A.4. Inference Time

In this section, we compare the inference times of our proposed attention-based architecture and the BGNN model. Specifically, we measure per-instance inference time on the four binary variable prediction tasks, reporting the average over the test instances in Table 13. We can see that although introducing attention increases the model's complexity, the increased latency is modest in practice.

*Table 13.* Average inference time (s) on four testing datasets.

| Method | CA | IS | IP | WA |
|--------|------|------|------|------|
| BGNN | 0.005 | 0.004 | 0.003 | 0.009 |
| Ours | 0.017 | 0.010 | 0.007 | 0.020 |

## B. Raw Features

The raw features of variables and constraints used in this paper align closely with the baseline representation in each predictive task. Specifically, the raw features used for instance-level, element-level and solving state-level prediction tasks follow Chen et al. (Chen et al., 2023), Han et al. (Han et al., 2023), and Gasse et al. (Gasse et al., 2019), as listed in Table 14, 15 and 16, respectively.

*Table 14.* Raw features for instance-level prediction

| ID | Variable Feature Name | Description |
|----|-----------------------|-------------|
| 0 | Objective | Objective function coefficient |
| 1 | Variable type | Indicates whether the variable is a binary variable |
| 2 | Lower bound | Lower bound of the variable |
| 3 | Upper bound | Upper bound of the variable |

| ID | Constraint Feature Name | Description |
|----|-------------------------|-------------|
| 0 | Right-hand constant | Constraint on the right-hand side constant |
| 1 | Constraint type | The sense of the constraint |

*Table 15.* Raw features for element-level prediction

| ID | Variable Feature Name | Description |
|------|------------------------------|-------------|
| 0 | Objective function coefficients | Indicates the weight of this variable in the objective function |
| 1 | Variable coefficient | The mean coefficient of the variable across all constraints |
| 2 | Variable degree | Degree of a variable node in the bipartite graph |
| 3 | Maximum variable coefficient | The maximum coefficient value of the variable across all constraints |
| 4 | Minimum variable coefficient | The minimum coefficient value of the variable across all constraints |
| 5 | Variable type | Indicates whether the variable is a binary variable |
| 6-17 | Position embedding | Encoding binary variable order (only effective on IP) |

| ID | Constraint Feature Name | Description |
|----|-------------------------|-------------|
| 0 | Constraint coefficient | The mean value of the constraint coefficients |
| 1 | Constraint degree | The number of variables involved in the constraints |
| 2 | Right-hand constant | Constraint on the right-hand side constant |
| 3 | Constraint type | The sense of the constraint |

## C. Description of Benchmark Problems

### C.1. Instance-Level Prediction

For instance-level prediction tasks, we follow the data generation protocol in Chen et al. (Chen et al., 2023) to generate instances with $n = 20$ variables and $m = 6$ constraints. Specifically, the procedure of generating unfoldable instances is as follows:

- The objective function coefficient $c_i$ of each variable $x_i$ is sampled from a normal distribution $\mathcal{N}(0, 0.01)$;

- The lower bound $l_i$ and upper bound $u_i$ of each variable $x_i$ are independently sampled from a normal distribution $\mathcal{N}(0, 10)$. If $l_i$ is sampled greater than $u_i$, then $l_i$ and $u_i$ are swapped to ensure $l_i \leq u_i$;

- Each variable $x_i$ has a 50% probability of being assigned as an integer variable; otherwise, it is a continuous variable;

- The right-hand side constant term $b_j$ of each constraint is sampled from a standard normal distribution $\mathcal{N}(0, 1)$;

- The constraint type is uniformly selected from the set $\{\leq, =, \geq\}$;

*Table 16.* Raw features for solving state-level prediction

| ID | Variable Feature Name | Description |
|---|---|---|
| 0-3 | Variable type | Employing one-hot encoding to denote whether a variable type is binary, integer, impl.integer, or continuous |
| 4 | Objective coefficient | Normalized objective function coefficients |
| 5 | Lower bound | Indicates whether there is a lower bound |
| 6 | Upper bound | Indicates whether there is an upper bound |
| 7 | At the lower bound | Indicates whether the current solution value equals the lower bound |
| 8 | At the upper bound | Indicates whether the current solution value equals the upper bound |
| 9 | Solution value fractionality | Indicates the extent of deviation from an integer |
| 10-13 | Basis status | Representing the basis states of the simplex method (lower, basic, upper, zero) using one-hot encoding. |
| 14 | Reduced cost | Marginal cost of non-basic variables. |
| 15 | Age | Indicates how many times the variable (or node) has undergone LP solving |
| 16 | Relaxation value | The solution value obtained after solving the LP relaxation problem for the variable at the current node |
| 17 | Optimal integer solution value | The values assumed by variables in the currently found optimal integer feasible solution |
| 18 | Average integer solution value | Average of historical integer solutions |

| ID | Constraint Feature Name | Description |
|---|---|---|
| 0 | Cosine similarity | Cosine similarity with respect to the objective function |
| 1 | Right-hand constant | Normalized constraint right-hand side constant |
| 2 | Tight | Indicates whether the constraint is tight |
| 3 | Dual solution value | Normalized dual solution value |

- The adjacency matrix $\mathbf{A}$ is initialized as a $6 \times 20$ zero matrix, with 60 positions randomly selected as nonzero elements. Each nonzero element $\mathbf{A}_{ij}$ is sampled from a standard normal distribution $\mathcal{N}(0, 1)$.

The foldable instances are generated in pairs. Specifically, we generate $k$ pairs of instances where $1 \leq k \leq 500$, and the procedure of generating each foldable instance is as follows:

- The objective function coefficient $c_i$ of each variable $x_i$ is set to 0;

- A subset $J = \{x_{j1}, x_{j2}, ..., x_{j6}\}$ of 6 variables is randomly selected from the 20 variables;

- Each $x_{ji} \in J$ is set to be a binary variable;

- Each $x_{ji} \notin J$ is set to be a continuous variable, with $l_i$ and $u_i$ generated the same as unfoldable instances;

- The $(2k-1)$-th instance is feasible, and its constraints form a hexagonal loop: $x_{j1}+x_{j2} = 1, x_{j2}+x_{j3} = 1, x_{j3}+x_{j4} = 1, x_{j4} + x_{j5} = 1, x_{j5} + x_{j6} = 1, x_{j6} + x_{j1} = 1$;

- The $2k$-th instance is infeasible, and its constraints form two triangular cycles: $x_{j1}+x_{j2} = 1, x_{j2}+x_{j3} = 1, x_{j3}+x_{j1} = 1, x_{j4} + x_{j5} = 1, x_{j5} + x_{j6} = 1, x_{j6} + x_{j4} = 1$.

## C.2. Element-Level Prediction

In element-level prediction tasks, we provide problem descriptions for the Maximum Independent Set (MIS), Combinatorial Auction (CA), Balanced Item Placement (IP), and Workload Appointment problems (WA). For the former two, instances are generated using the Ecole library following the methodology of Gasse et al. (Gasse et al., 2019). Instances for the latter two can be directly downloaded from the ML4CO competition website (Gasse et al., 2022). For all instances, we utilise 400 for training and 100 for testing. Additionally, for both the IP and WA datasets, we train on instances indexed from 0 to 399 and test on instances indexed from 9900 to 9999. Below we outline the generative models for each problem and the specific generative parameters for the MIS and CA problems.

### C.2.1. MAXIMUM INDEPENDENT SET

In the MIS problem, given an undirected graph $G \equiv (V, E)$, the objective is to select a maximum subset $V' \subset V$ such that no two nodes are adjacent, i.e., no edge connects them. The generative model for MIS problems (Bergman et al., 2016) is formulated as the following mixed-integer linear programming problem:

$$
\begin{aligned}
\max_{x} \quad & \sum_{v \in V} x_v \\
\text{s.t.} \quad & x_u + x_v \leq 1 \quad \forall (u, v) \in E \\
& x_v \in \{0, 1\} \quad \forall v \in V
\end{aligned}
$$

The binary variable $x_v$ indicates whether node $v$ is selected into the independent set (1 denotes selected, 0 denotes not selected). The constraints ensure that for each edge $(u, v)$, nodes $u$ and $v$ cannot be selected simultaneously. We generated instances that are as challenging as possible for our hardware. We set the number of nodes to $n = 3000$, and the edge existence probability to $0.1$.

### C.2.2. COMBINATORIAL AUCTION

In the CA problem, given $m$ items, there are $n$ bids $\{(B_i, p_i) : i \in [n]\}$, where $B_i$ denotes a subset of items and $p_i$ is the price for that combination. The objective is to assign items to bids to maximize revenue, with each item being assigned at most once. The MILP model for CA (Leyton-Brown et al., 2000) is formulated as:

$$
\begin{aligned}
\max_{x} \quad & \sum_{i=1}^{n} p_i x_i \\
\text{s.t.} \quad & \sum_{i: j \in B_i} x_i \leq 1 \quad \forall j \in [m] \\
& x_i \in \{0, 1\} \quad \forall i \in [n]
\end{aligned}
$$

The binary variable $x_i$ indicates whether bid $i$ is accepted (1 for acceptance and 0 otherwise). The constraints ensure that each item $j$ is included in at most one accepted bid. We generate CA instances with $m = 8000$ items and $n = 3000$ bids, with bid prices bounded in [1,100].

### C.2.3. BALANCED ITEM PLACEMENT

The IP problem simulates the placement of items (such as data files or computational processes) within containers (such as hard disks or servers) in a large-scale distributed system, such as a data centre. The objective is to achieve the most balanced utilization of containers, whilst satisfying various capacity constraints, thereby avoiding hotspots or resource waste. The problem also incorporates a practical constraint: the system currently possesses an existing placement scheme, but permits re-optimization of this placement within a finite relocation cost (i.e., a limit on the maximum number of items that may be moved). An IP instance comprises three sets: the item set $I$, the container set $J$, and the dimension set $K$ (e.g., CPU, memory, disk I/O, etc.). Then the MILP formulation of IP is:

$$
\begin{aligned}
\min_{x,y,z} \quad & \sum_{j \in J} \sum_{k \in K} \alpha_k y_{jk} + \sum_{k \in K} \beta_k z_k \\
\text{s.t.} \quad & \sum_{j \in J} x_{ij} = 1 & \forall i \in I \\
& \sum_{i \in I} a_{ik} x_{ij} \leq b_k & \forall j \in J, \forall k \in K \\
& \sum_{i \in I} d_{ik} x_{ij} + y_{jk} \geq 1 & \forall j \in J, \forall k \in K \\
& y_{jk} \leq z_k & \forall j \in J, \forall k \in K \\
& x_{ij} \in \{0, 1\} & \forall i \in I, \forall j \in J \\
& y_{jk} \geq 0 & \forall j \in J, \forall k \in K
\end{aligned}
$$

The binary variable $x_{ij}$ indicates whether item $i$ is placed in container $j$ (1 for true and 0 for false). The parameter $a_{ik}$ denotes the size or demand of item $i$ along dimension $k$, while $b_k$ represents the capacity limit for each container in dimension $k$. The parameter $d_{ik}$ is the imbalance contribution coefficient of item $i$ on dimension $k$. The continuous, non-negative variable $y_{jk}$ measures the extent to which the total imbalance contribution of items in container $j$ for dimension $k$ falls below a predefined threshold (value of 1 in the constraint $\sum_i d_{ik} x_{ij} + y_{jk} \geq 1$). Finally, the continuous variable $z_k$ represents the maximum value of $y_{jk}$ across all containers for dimension $k$. The objective function minimizes a weighted sum of these maximum imbalance measures across all dimensions.

### C.2.4. WORKLOAD APPOINTMENT

The WA problem simulates the allocation of workloads (such as data stream processing tasks) to the minimum number of workers (such as servers). A core requirement is allocation robustness: the entire system must continue functioning normally even if any single worker fails. No tasks should be lost. This is typically achieved through redundancy, such as ensuring each task can be handled by multiple workers. The problem is modelled as a packing problem with allocation constraints. Given the task set $M$, the worker set $N$, and the subset $N_i$ of workers capable of handling task $i$, the standard MILP formulation of WA is as follows:

$$
\begin{aligned}
\min_{x,y} \quad & \sum_{j \in N} y_j \\
\text{s.t.} \quad & x_{ij} \leq a_i y_j && \forall i \in M, \forall j \in N^i \\
& \sum_{i \in M : j \in N^i} x_{ij} \leq b_j && \forall j \in N \\
& \sum_{j \in N^i \setminus \{j'\}} x_{ij} \geq a_i && \forall i \in M, \forall j' \in N^i \\
& y_j \in \{0,1\} && \forall j \in N \\
& 0 \leq x_{ij} \leq b_j && \forall i \in M, \forall j \in N^i
\end{aligned}
$$

The binary variable $y_j \in \{0,1\}$ indicates whether worker $j$ is activated (1 for yes, 0 for no). The continuous variable $x_{ij}$ denotes the proportion of task $i$ allocated to worker $j$. $a_i$ represents the load size of task $i$, while $b_j$ denotes the capacity of worker $j$.

## C.3. Solving State-Level Prediction

This section introduces the benchmark problems for solving state-level prediction tasks, including Set Covering (SC), Combinatorial Auctions (CA), Capacitated Facility Location with Unsplittable Demand (CFL), and Maximum Independent Set (MIS). The description of CA and MIS are the same as in the element-level prediction tasks, while the introduction of SC (Balas & Ho, 2009) and CFL (Cornuéjols et al., 1991) are given below.

### C.3.1. SET COVERING

In the SC problem, given $m$ elements and $n$ sets of elements, we aim to cover all elements using as few sets as possible. The union of $n$ sets forms a collection $S$ covering $m$ elements, each of which belongs to at least one set. The SC problem can be formulated as follows (Balas & Ho, 2009):

$$
\begin{aligned}
\min \quad & \sum_{i=1}^{n} x_i \\
\text{s.t.} \quad & \sum_{i : j \in s_i} x_i \geq 1, && j = 1, \ldots, m, \\
& x_i \in \{0,1\}, && i = 1, \ldots, n.
\end{aligned}
$$

where $x_i$ denotes whether the $i$-th set $s_i$ is selected. For the SC problem, we train and test on instances with $m = 400, n = 750$, and transfer on instances with $m = 500, n = 1000$.

### C.3.2. COMBINATORIAL AUCTIONS

In this part of experiments, we train and test on instances with $m = 100$ items and $n = 500$ bids, and the transfer experiment is on instances with $m = 200$ items and $n = 1000$ bids.

### C.3.3. CAPACITATED FACILITY LOCATION WITH UNSPLITTABLE DEMAND

In the CFL problem, given $n$ customers with demand $\{d_j\}_{j=1}^{n}$ and $m$ facilities with fixed operating costs $\{f_i\}_{i=1}^{m}$ and capacity $\{s_i\}_{i=1}^{m}$, let $c_{ij}/d_j$ be the unit transportation cost between facility $i$ and customer $j$, and $p_{ij}/d_j$ be the unit profit

generated by facility $i$ supplying customer $j$. The CFL problem is formulated as follows (Cornuéjols et al., 1991):

$$\min \sum_{i=1}^{m} \sum_{j=1}^{n} c_{ij} x_{ij} + \sum_{i=1}^{m} f_i y_i$$

$$\text{s.t.} \sum_{j=1}^{n} d_j x_{ij} \le s_i y_i, \quad i = 1, ..., m$$

$$\sum_{i=1}^{m} x_{ij} \ge 1, \quad j = 1, ..., n$$

$$x_{ij} \in \{0, 1\} \quad \forall i, j$$

$$y_i \in \{0, 1\} \quad \forall i$$

where each variable $x_{ij}$ denotes the decision of facility $i$ to meet the demand of customer $j$, while each variable $y_i$ denotes whether facility $i$ is operated. For the CFL problem, we train and test on instances with $n = 35$, $m = 35$, and transfer on instances with $n = 60$, $m = 35$.

### C.3.4. MAXIMUM INDEPENDENT SET

In the MIS problem, we train and test on instances with $n = 500$, and transfer on instances with $n = 1000$. The probability of an edge existing between any two nodes in the graph is set to 0.25.

## D. Primal Gap and Primal Integral

Here we give the definition of the Primal Gap (PG) and Primal Integral (PI) metrics following (Berthold, 2013). PG is used to measure the relative distance between the objective value $\mathbf{c}^\top \tilde{\mathbf{x}}$ of a feasible solution $\tilde{\mathbf{x}}$ and the best known objective value (BKV), denoted by $\mathbf{c}^\top \mathbf{x}^*$. It is a scalar value in [0,1], where a smaller value indicates the solution is closer to optimal. PG is defined as:

$$PG(\tilde{\mathbf{x}}) = \begin{cases} 0, & \text{if } |\mathbf{c}^\top \tilde{\mathbf{x}}| = |\mathbf{c}^\top \mathbf{x}^*| = 0, \\ 1, & \text{if } \mathbf{c}^\top \mathbf{x}^* \cdot \mathbf{c}^\top \tilde{\mathbf{x}} < 0, \\ \frac{|\mathbf{c}^\top \mathbf{x}^* - \mathbf{c}^\top \tilde{\mathbf{x}}|}{\max\{|\mathbf{c}^\top \mathbf{x}^*|, |\mathbf{c}^\top \tilde{\mathbf{x}}|\}}, & \text{else.} \end{cases}$$

Note that for two feasible solutions $\tilde{\mathbf{x}}_1$ and $\tilde{\mathbf{x}}_2$, if $\mathbf{c}^\top \tilde{\mathbf{x}}_1 < \mathbf{c}^\top \tilde{\mathbf{x}}_2$ and $\text{sgn}(\mathbf{c}^\top \tilde{\mathbf{x}}_1) = \text{sgn}(\mathbf{c}^\top \tilde{\mathbf{x}}_2)$, then $PG(\tilde{\mathbf{x}}_1) < PG(\tilde{\mathbf{x}}_2)$.

Based on PG, we can define a time-varying PG function $p(t)$ as:

$$p(t) = \begin{cases} 1, & \text{if no incumbent found by time } t \\ PG(\tilde{\mathbf{x}}(t)), & \text{where } \tilde{\mathbf{x}}(t) \text{ denotes the incumbent at time } t \end{cases}$$

which enables us to define PI as the integral of the PG function $p(t)$ over the whole solving time $T$, serving to measure the cumulative performance of the entire solving process in terms of solution quality. It reflects the relationship between obtaining high-quality solutions and time expenditure, with lower values indicating higher efficiency of the solving process (Berthold, 2013). Specifically, PI is defined as follows:

$$PI(T) = \int_{t=0}^{T} p(t) dt = \sum_{i=1}^{I} p(t_{i-1}) \cdot (t_i - t_{i-1})$$

where $t_i$ denotes the time at which the $i$-th incumbent is found with $t_0 = 0$, $t_I = T$.

## E. Implementation Details

### E.1. Hyperparameters of Model Training

As mentioned in Section 4, throughout the experiments, we use the same neural architecture for both BGNN (2 layers) and our method (4 layers and 2 heads) to extract 64-dimensional features for both variables and constraints. Other training-related hyperparameters are reported in Table 17.

*Table 17.* Hyperparameters of the BGNN and our neural architecture for different tasks

| Task | Model | Batch size | Learning rate | Epochs |
|---|---|---|---|---|
| Feasibility prediction | BGNN | 64 | 3e-4 | 10000 |
|  | Ours | 64 | 8e-4 | 10000 |
| Optimal objective value prediction | BGNN | 64 | 3e-4 | 12000 |
|  | Ours | 64 | 8e-4 | 12000 |
| Binary variable prediction | BGNN | 64 | 3e-3 | 100 |
|  | Ours | 64 | 8e-4 | 100 |
| Branching strategy learning | BGNN | 64 | 1e-3 | 1000 |
|  | Ours | 64 | 1e-3 | 1000 |

We further examine the sensitivity of our model to the number of layers $T$ and hidden dimension $d$ on WA. As shown in Table 18, increasing $T$ and $d$ generally improves MCC. This is consistent with Fig. 3h, which shows that our model benefits from increasing depth in terms of Macro-F1. While it is possible to obtain better results with larger $T$ and $d$, we use $T = 4$ and $d = 64$ as a reasonable trade-off between prediction accuracy and efficiency. This setting also leads to robustly good performance in other datasets.

*Table 18.* Sensitivity of MCC to the number of layers $T$ and hidden dimension $d$ on WA.

|  | $d = 16$ | $d = 32$ | $d = 64$ | $d = 128$ | $d = 256$ |
|---|---|---|---|---|---|
| $T = 1$ | 0.7121 | 0.7303 | 0.7307 | 0.7316 | 0.7321 |
| $T = 2$ | 0.7264 | 0.7418 | 0.7568 | 0.7640 | 0.8035 |
| $T = 3$ | 0.8025 | 0.8097 | 0.8312 | 0.8415 | 0.8387 |
| $T = 4$ | 0.8125 | 0.8157 | 0.8415 | 0.8450 | 0.8489 |
| $T = 5$ | 0.8195 | 0.8271 | **0.8476** | 0.8482 | **0.8518** |

## E.2. Hyperparameters of Predict-and-Search Frameworks

For predict-and-search experiments in Section 4.2 and 4.5, the search-related hyperparameters fine-tuned for each method and dataset are listed in Table 19 and 20. Within the PaS framework, enhanced prediction accuracy enables us to confidently fix more binary variables to achieve superior results. Conversely, under the Apollo framework, where variable fixing is required to correct actual prediction outcomes, we employ identical parameters to BGNN.

*Table 19.* Hyperparameters of the PaS framework. Each entry denotes $(k_0, k_1, \Delta)$

| Method | IP | WA | MIS | CA |
|---|---|---|---|---|
| BGNN | $(400, 5, 40)$ | $(0, 500, 40)$ | $(300, 300, 40)$ | $(40, 0, 40)$ |
| Ours | $(400, 5, 40)$ | $(0, 500, 40)$ | $(400, 400, 40)$ | $(50, 0, 40)$ |
| OPTFM | $(200, 5, 40)$ | $(0, 500, 40)$ | $(300, 300, 40)$ | $(50, 0, 40)$ |
| Polynormer | $(400, 5, 40)$ | $(0, 300, 40)$ | $(200, 200, 40)$ | $(50, 0, 40)$ |
| MAGCN | $(400, 5, 40)$ | $(0, 500, 40)$ | $(400, 400, 40)$ | $(50, 0, 40)$ |
| GCON | $(200, 5, 40)$ | $(0, 200, 40)$ | $(300, 300, 40)$ | $(50, 0, 40)$ |
| GNN-SSM | $(200, 5, 40)$ | $(0, 500, 40)$ | $(300, 300, 40)$ | $(50, 0, 40)$ |

*Table 20.* Hyperparameters of the Apollo framework (values within each parentheses is the combination $(k_0^{(i)}, k_1^{(i)}, \Delta^{(i)})$ of iteration $i$)

| Iteration | Dataset | | | |
|---|---|---|---|---|
|  | IP | WA | MIS | CA |
| 1 | (100,20,50) | (60,600,5) | (50,50,50) | (100,0,60) |
| 2 | (40,15,20) | (50,500,5) | (40,15,40) | (50,0,50) |
| 3 | (20,15,10) | (40,400,5) | (20,15,30) | (40,0,40) |
| 4 | (5,50,30) | (30,0,5) | (1,5,10) | (30,0,30) |

## E.3. Implementation Environment

Our model was implemented using PyTorch 2.4.1 and trained on an Ubuntu 20.04.6 LTS system equipped with an Intel(R) Core(TM) i9-10920X CPU operating at 3.5 GHz, 128 GB of memory, and an NVIDIA GeForce RTX 4090 GPU.

### E.4. Configuration of Advanced Architectures

For the experiments in Section 4.5, we tune the depth of each architecture on the WA dataset and use the selected configuration in the main comparison for fairness. As shown in Table 21, OPTFM achieves the highest MCC with 1 layer, while MAGCN, GCON, and GNN-SSM achieve the highest MCC with 2 layers.

*Table 21.* Depth selection for advanced architectures on WA (values are MCC).

| Method | 1 layer | 2 layers | 3 layers | 4 layers | 5 layers |
|---|---|---|---|---|---|
| OPTFM | **0.7155** | 0.6858 | 0.6466 | 0.5532 | 0.5206 |
| MAGCN | 0.7583 | **0.7971** | 0.7939 | 0.7963 | 0.7961 |
| GCON | 0.7465 | **0.7991** | 0.7970 | 0.7921 | 0.7914 |
| GNN-SSM | 0.7598 | **0.8022** | 0.8006 | 0.8021 | 0.7986 |

For Polynormer, we tune both local and global depths. As shown in Table 22, the highest MCC is obtained with 3 local layers and 2 global layers. Based on these results, we use 1-layer OPTFM, 2-layer MAGCN/GCON/GNN-SSM, and Polynormer with 3 local and 2 global layers in the main comparison in Section 4.5.

*Table 22.* Depth selection for Polynormer on WA (values are MCC) .

| Local depth | Global depth | | |
|---|---|---|---|
| | 1 | 2 | 3 |
| 1 | 0.7460 | 0.7455 | 0.7427 |
| 2 | 0.7431 | 0.7408 | 0.7463 |
| 3 | 0.7412 | **0.7498** | 0.7372 |

