# OpenReview forum: "A General Neural Backbone for Mixed-Integer Linear Optimization via Dual Attention"
_ICML.cc/2026/Conference — ICML 2026 regular_

### Official Review · Reviewer_AP6h · 2026-03-05

**Soundness:** 2
**Presentation:** 2
**Significance:** 2
**Originality:** 1
**Overall Recommendation:** 3
**Confidence:** 5

**Summary:**

This paper proposes an attention-driven neural backbone with a dual attention mechanism for MILP, overcoming GNNs' local representation limits by capturing global and long-range dependencies. The authors validate the model on three key MILP tasks, and it consistently outperforms traditional GNN architectures, proving attention-based modeling's value for combinatorial optimization.

**Compliance With Llm Reviewing Policy:**

Affirmed.

**Final Justification:**

The authors have addressed my most concerns, but the authos still lack the expressive power of the new GNN architecture. Therefore, it is difficult to determine how much of the performance boost came from the model versus the fine-tuning hyperparameters. So, I maintain my score.

**Key Questions For Authors:**

1. The literature review is incomplete and fails to cover the latest relevant works, such as DiffILO (ICLR 2025) and CoCo-MILP (AAAI 2026).
2. The authors should apply the dual attention mechanism to more baselines to demonstrate its generalizability, and report detailed experimental settings. For instance, the impact of hyperparameters like k0 and k1 on model performance in the predict-and-search framework.
3. It is necessary to conduct a thorough comparison between the proposed dual attention mechanism and SOTA graph transformer architectures, as well as other novel graph representation learning architectures.

**Limitations:**

yes

**Strengths And Weaknesses:**

The authors argue that GNNs, which are commonly used to represent MILPs as bipartite graphs, suffer from limited expressive power, and thus design a new GNN architecture integrated with a dual attention mechanism. However, this idea lacks significant novelty, as it essentially mimics recent advances in the GNN field to construct a bipartite graph transformer for MILPs, similar to Graphormer. Additionally, the model has a relatively high time complexity, and the baselines adopted in this paper are outdated, making it necessary to incorporate more SOTA baselines for comparison.

---

> ### Author Rebuttal · Authors · 2026-03-31
>
> Thank you so much for the valuable comments. All rebuttal tables are available at https://anonymous.4open.science/r/DAB/tables.pdf
>
> **W1 Novelty**
>
> We would like to respectfully clarify that our contribution is *not* a bipartite graph transformer (GT). Rather, our contribution is a new information flow design tailored to MILP. In particular, prior BGNN-style architectures follow a sequential v→c→v message passing paradigm, where constraint nodes act as an intermediate bottleneck for information propagation. In contrast, our backbone performs self-attention within variable and constraint nodes together with cross-attention between the two sets *in parallel*, allowing direct and simultaneous interaction between variables and constraints. This removes the intermediate compression inherent in two-stage message passing and enables the model to capture both local and global structural patterns more effectively.
>
> As shown in Table 7 in our paper and Table R15-R17 in the response, simply applying GT such as Polynormer [R2] in a bipartite fashion does not significantly enhance performance. We did not use Graphormer since its full attention is not practical for large MILP graphs. Polynormer is more scalable and up-to-date, making it a fairer comparator. Even against this strong GT baseline, our method remains competitive, supporting that it is not simply a bipartite reimplementation of GT.
>
> **W2 Efficiency**
>
> The efficiency of our model is practical. As analyzed in Sec. 3.1, our self-attention is linearized and the cross-attention is sparse over the nonzero pattern of the MILP matrix, giving a per-layer complexity of $O((m+n+e)d^2)$ time and $O((m+n+e)d)$ memory, the same asymptotic complexity as BGNN. Practically, our method scales to real-world MILP instances up to 550k variables and 1.1M nonzeros as in Appendix A.3 on a single GPU, showing the practical scalability of our method.
>
> **W3&Q3 Stronger baselines**
>
> We expand the comparison to OPTFM [R1], Polynormer [R2], MAGCN [R3], GCON [R4], and GNN-SSM [R5]. OPTFM and MAGCN are developed for MILP, and are therefore closely related to our work. Polynormer, GCON, and GNN-SSM are developed for homogeneous graphs, hence we modify them in a bipartite v→c→v fashion. Due to rebuttal time constraints, we restrict experiments to the four element-level tasks (binary variable prediction). We use 1-layer OPTFM, 2-layer MAGCN/GCON/GNN-SSM, and Polynormer with 3 global and 2 local layers, which performs the best as in Table R6-R8.
>
> As shown in Table R15–R17, adding these stronger baselines does not change the overall picture. Across the four datasets (Table R15), our method is the strongest overall, ranking first on 13 out of the 16 metric-dataset pairs. It is consistently best on WA, MIS, and CA across all four metrics, and remains competitive on IP. We note that IP is a more special case with strong symmetry and a highly imbalanced label distribution, so ErrorRate alone can be less informative. For instance, OPTFM has the lowest ErrorRate but its MCC is nearly zero. For the downstream solving performance (Tables R16 and R17), our method also remains highly competitive.
>
> **Q1 Related works**
>
> Thank you for pointing out these works. DiffILO studies unsupervised differentiable learning for ILP with BGNN, while CoCo-MILP is developed within the PaS framework and improves prediction via contrastive learning and an intra-constraint competitive GNN. By contrast, our method is intended as a more general-purpose backbone for MILP learning across multiple tasks. We will explicitly discuss these related works in the revision.
>
> **Q2 Baselines and hyperparameter**
>
> For more baselines, please refer to our response to **W3&Q3**.
>
> Due to rebuttal time constraints, we analyze the impact of PaS hyperparameters $k_0$ and $k_1$ using our model on WA. As shown in Table R13-R14, PaS performance is sensitive to $k_0$ and $k_1$, with both PG and PI varying noticeably across different (k0,k1). This sensitivity is also known in other works such as DiffILO and ConPas.
>
> ref:
>
> [R1] OPTFM: A Scalable Multi-View Graph Transformer for Hierarchical Pre-Training in Combinatorial Optimization. NeurIPS 2025
>
> [R2] Polynormer: Polynomial-Expressive Graph Transformer in Linear Time. ICLR 2024
>
> [R3] Graph convolutional network with multiple aggregators for learning to branch. Knowledge-Based Systems 2025
>
> [R4] Towards a General Recipe for Combinatorial Optimization With Multi-Filter GNNs. Learning on Graphs (LoG) 2024
>
> [R5] On Vanishing Gradients, Over-Smoothing, and Over-Squashing in GNNs: Bridging Recurrent and Graph Learning. NeurIPS 2025

---

> > ### Author Rebuttal · Reviewer_AP6h · 2026-04-03
> >
> > Thank you for your response.
> >
> > First, after reviewing the extensive results, could you please report the improvement in terms of the objective function values? To my understanding, in MILP problems, a high accuracy in binary variable prediction does not necessarily guarantee superior architectural performance. Sometimes, the prediction of critical variables, such as backdoor variables, is more decisive.
> >
> > Second, I still have concerns regarding the contribution of this paper. It appears to be merely adding attention mechanisms to a previous architecture. The authors should provide theoretical justification for the improvement in expressive power.

---

> > > ### Author Response · Authors · 2026-04-04
> > >
> > > Thank you for the followup questions.
> > >
> > > **Q1.** Table 3 in our paper reported the PG-based improvement, which is consistent with the objective-gap improvement in PaS/Apollo/CoCo-MILP. Here we update Table R16 with objective value comparison with average PG improvement relative to BGNN. Results show that our method delivers the highest improvement over BGNN. The average PI improvement in Table R17 further shows the stronger ability in converging to optimality.
> > >
> > > Indeed, a high accuracy in variable prediction does not necessarily translate to better downstream solving performance. However, we would like to respectfully clarify that this does not preclude the necessity of a high-precision prediction model. Intuitively, high-quality predictions are critical for prediction-based MILP methods, and in general, more accurate predictions provide more reliable solving guidance. The prediction of important variables such as backdoors are certainly of interest, but the current scope of our work is to design a neural backbone for general MILP representation learning, instead of focusing on binary variable prediction. The key advantage of our method is the ability in extracting deeper representations and capturing long-range dependencies, which can also benefit backdoor learning methods and is certainly an interesting future work.
> > >
> > > **Q2.** We would like to respectfully clarify that our contribution is not simply adding attention modules to a previous architecture. Rather, we redesign the information flow on MILP representation learning. Standard BGNN-style methods propagate information through a sequential two-stage paradigm (v-to-c and c-to-v). In contrast, our backbone jointly models variable-variable, constraint-constraint, and variable-constraint interactions through a unified dual-attention, and the four modules work *in parallel*. To the best of our knowledge, such architecture does not exist in the literature, and even the very recent OPTFM still follows the BGNN-style paradigm.
> > >
> > > Here we give a more principled justification. The matrices below are not explicit parameter matrices in implementation, but effective local operators used to compare interaction structures of BGNN and our model. A BGNN layer is executed sequentially to first update constraints representations, and then update variables representations:
> > >
> > > $$
> > > \\begin{bmatrix}
> > >  \\mathbf{H}^{V}\_{t-1}\\\\
> > >  \\widetilde{\\mathbf{H}}^{C}\_{t}
> > > \\end{bmatrix}
> > > \=
> > > \\begin{bmatrix}
> > > I & 0\\\\
> > > P\_{cv,t} & S\_{c,t}
> > > \\end{bmatrix}
> > > \\begin{bmatrix}
> > >  \\mathbf{H}^{V}\_{t-1}\\\\
> > >  \\mathbf{H}^{C}\_{t-1}
> > > \\end{bmatrix},
> > > \\qquad
> > > \\begin{bmatrix}
> > >  \\mathbf{H}^{V}\_{t}\\\\
> > >  \\mathbf{H}^{C}\_{t}
> > > \\end{bmatrix}
> > > \=
> > > \\begin{bmatrix}
> > > S\_{v,t} & P\_{vc,t}\\\\
> > > 0 & I
> > > \\end{bmatrix}
> > > \\begin{bmatrix}
> > >  \\mathbf{H}^{V}\_{t-1}\\\\
> > >  \\widetilde{\\mathbf{H}}^{C}\_{t}
> > > \\end{bmatrix},
> > > $$
> > >
> > > where $P_{cv,t}$ and $P_{vc,t}$ denote cross-type propagation operators at layer $t$ transfer information along bipartite edges from variables to constraints and from constraints to variables, respectively. $S_{c,t}$ and $S_{v,t}$ denote self-update operators for contributions of previous representations themselves. Composing the two stages gives
> > >
> > > $$
> > > \\begin{bmatrix}
> > >  \\mathbf{H}^{V}\_{t}\\\\
> > >  \\mathbf{H}^{C}\_{t}
> > > \\end{bmatrix}
> > > \=
> > > \\begin{bmatrix}
> > > S\_{v,t}+P\_{vc,t}P\_{cv,t} & P\_{vc,t}S\_{c,t}\\\\
> > > P\_{cv,t} & S\_{c,t}
> > > \\end{bmatrix}
> > > \\begin{bmatrix}
> > >  \\mathbf{H}^{V}\_{t-1}\\\\
> > >  \\mathbf{H}^{C}\_{t-1}
> > > \\end{bmatrix}.
> > > $$
> > >
> > > By contrast, our dual-attention backbone is a four-block interaction structure:
> > >
> > > $$
> > > \\begin{bmatrix}
> > >  \\mathbf{H}^{V}\_{t}\\\\
> > >  \\mathbf{H}^{C}\_{t}
> > > \\end{bmatrix}
> > > \=
> > > \\begin{bmatrix}
> > > A\_{vv,t} & A\_{vc,t}\\\\
> > > A\_{cv,t} & A\_{cc,t}
> > > \\end{bmatrix}
> > > \\begin{bmatrix}
> > >  \\mathbf{H}^{V}\_{t-1}\\\\
> > >  \\mathbf{H}^{C}\_{t-1}
> > > \\end{bmatrix},
> > > $$
> > >
> > > where $A_{vv,t}, A_{vc,t}, A_{cv,t}, A_{cc,t}$ denote interaction blocks induced by our dual-attention layer corresponding to v->v, c->v, v->c, and c->c interactions, respectively.
> > >
> > > Under sequential two-stage propagation, the four blocks are coupled through cross-type propagation and self-update. By contrast, our dual-attention induces direct same-type and cross-type interactions within one layer, leading to a less constrained interaction structure. Thus, under this simplified operator view, the two-stage update is a restricted case of our interaction structure.
> > >
> > > This distinction is also reflected empirically. In our experiments, both BGNN and the staged transformer-style baseline OPTFM underperform our model. The depth behavior on WA is markedly different: OPTFM drops from 0.7155 to 0.5206 as layers increase from 1 to 5, while our model rises from 0.7307 to 0.8476 at $d=64$ (Tables R7 and R1). This suggests that our parallel dual-attention is more effective and stable than sequential two-stage propagation, even with Transformer-style attention.

---

### Official Review · Reviewer_bqNa · 2026-03-13

**Soundness:** 3
**Presentation:** 3
**Significance:** 3
**Originality:** 2
**Overall Recommendation:** 5
**Confidence:** 3

**Summary:**

The authors propose a novel neural architecture for solving mixed-integer linear optimization problems (MILP) through their bipartite graph representations. While prior bipartite GNN models are typically limited in depth due to rapid over-smoothing and over-squashing, the proposed model leverages self-attention within constraint and variable nodes each, and bidirectional cross-attention across them to alleviate these issues and discover non-local, long-range relationships over MILP graphs more effectively. The proposed architecture attains convincing results on instance, element and state-level prediction tasks, and is well-supported by additional studies on long-range dependency modelling and interpretability.

**Compliance With Llm Reviewing Policy:**

Affirmed.

**Final Justification:**

The authors propose a well-motivated method and an array of mechanistic analyses that make this an appealing work. The main drawback of the paper was convincing experimental results against up-to-date, competitive benchmarks, which the authors successfully alleviated in the rebuttal. I thus increase my score to 5, with the caveat that the authors comprehensively document the experimental settings and provide a reproducible codebase. I however reduce my confidence to 3, as my score increase partially hinges on the extensive evaluations added later on (and thus I have no way of verifying for correctness).

**Key Questions For Authors:**

1. Follow-up on Weakness 1.3: How does a 2-dual-attention-layer network do on Table 1? How would adjusting for similar parameter counts and/or depth affect the overall evaluation?
2. Follow-up on Weakness 2: Can you either empirically or theoretically characterize the superior expressivity of the proposed method over BGNN?
3. Have you tested the efficiency gains/performance losses from using a linearized approximation of self-attention? Do you have any idea on the viable scalability of the method? Even the linearized approximation could lead to significant slowdowns when instance sizes go in the thousands, and my understanding is that none of the benchmarks considered go up to such sizes. I do not pose this as a potential weakness, but as a point of curiosity and possible future work.
4. In Table 15 (Appendix E.2), why do the hyperparameters differ for BGNN vs the proposed method on MIS and CA? Wouldn’t it be simpler to use identical parameters as in Apollo for a fair comparison?

**Limitations:**

The authors have not explicitly discussed the limitations underlined in the Weaknesses section.

I think the core of the paper is sound, and the mechanistic analysis section in particular is worthy of praise. However, the baselines compared against are very limited as they’re allbased on a likely underparametrized BGNN model with lower receptive field, making a fair comparison difficult. That, and the oversimplified narrative put forth that does not acknowledge alternative ways of circumventing over-smoothing/-squashing in GNNs (and taking the superior expressivity of the proposed method for granted) hold the paper back and require some revision.

**Strengths And Weaknesses:**

**Stengths:**
1. The problem at hand and the proposed solution are well-motivated — the dated BGNN framework is typically taken for granted in tackling bipartite graphs with GNNs, with its inherent limitations that the authors address explicitly. The framing of the proposed method is a sound (e.g. through the use of linearized attention), novel (with some caveats — see weaknesses) and effective way of improving on said limitations.
2. The additional analyses in section 4.4 are very well-excecuted and are an excellent addition to the paper, providing interpretability and a mechanistic understanding of the advantages of the proposed method over conventional BGNNs.
3. The evaluation procedures are comprehensive. The authors test a large variety of interesting settings: instance, element and state-level tasks with several benchmarks per task, improving solver efficiency with PaS/Apollo, imitation learning for state-level tasks etc. The results show superiority over BGNN (though I remain unconvinced in certain aspects, see Weakness 1.3)

**Weaknesses:**
1. A major drawback of the paper is that the only baseline it is compared against is BGNN. BGNN is indeed popular to the point that it’s usually taken for granted as the de facto standard on bipartite CO graph representations; the authors largely follow this claim and treat BGNN as the only prior model of interest, whereas other approaches exist. This causes two set of issues:
   1. The proposed method lacks comparison with other published works on applying GNNs or graph transformers to bipartite CO graphs, both in related work and evaluations. Wang et al. (2023) [1] extends BGNN formulation with a Graph Pointer Network to improve performance, Vinh Toan and Van Hop (2025) [2] leverages PNA aggregation to improve expressivity, while Yuan et al. [3] propose a novel graph transformer architecture (also with self- and cross-attention modules for constraints and variables) for bipartite MILP settings. the former [1] doesn’t seem to have an available implementation, while the latter two [2, 3]  are quite recent (though published before the 2-month grace period) so that the authors may not be aware of them — but I would urge the authors to (a) clearly mention some of these works that they deem relevant in the appropriate section, and (2) at least attempt to benchmark one or two of these methods in a subset of their studies to provide a proper evaluation of their method in comparison to more recent work in the field.
   2. In relation to the aforementioned point, I find the narrative pushed forth by the authors that “GNNs will invariably suffer from over-smoothing and over-squashing” reductive. There is a *vast* literature available on alleviating over-smoothing and over-squashing in GNNs; methods such as graph rewiring strategies, virtual nodes, even basic residual/jumping knowledge connections are known to alleviate (a subset of) these issues and enable deep GNNs that achieve competitive results, many of which (not necessarily all, e.g. breaking the bipartite graph through rewiring may be undesirable) could viably be applied to BGNN-like methods. If one wants to keep the graph topology intact, there exist graph convolutions that inherently handle over-smoothing/-squashing, e.g. Wenkel et al. (2024) [4] propose GCON layers that attain excellent performance even when 20+ layers are stacked on relatively small CO graphs. I do not expect the authors to engage with *all* relevant literature, but moreso qualify their claims in a way that acknowledges the points above.
   3. To follow up, the authors take their claim that BGNNs heavily over-smooth/-squash for granted, and do not support this with any empirical evidence. I conceptually agree with their claim, but even so I find the argument that stacking more than two layers is harmful (up to five layers-deep GNNs are fairly common) surprising and requires backing. This is especially concerning since as per Appendix E.1, in all experiments, a BGNN with 2 layers is used whereas the proposed method uses 4 layers. I would like to see a fair comparison on both width and parameter counts (the 2-layer BGNN may be significantly underparametrized compared with a 4-layer proposed model with 2 attention heads for each set of attention computed) to be convinced of the superiority of the proposed method. As an example, the graphs used in Table 1 are tiny ($n = 20$ variables, $m=6$ constraints) — it is possible that the receptive field of 2 hops is not sufficient to cover the graph while 4 is, which could explain the 6% error rate from the Chen et al. (2023) paper.
2. The authors make multiple claims of higher expressivity over GNNs for MILPs (and cite several works on such theoretical studies), yet this is not supported beyond the experimental results that serve as a rough proxy of expressivity which may or may not hold (one can find many tasks a GCN outperforms a GIN despite the latter being provably more expressive, for example). I appreciate that the paper is not necessarily theoretically-driven and provable expressivity is not a main focus, but more concrete evidence towards this end would strengthen the paper considerably.
3. Minor typos (no effect on score):
   1. The “short” name of the paper in the header needs to be updated, still reads “Submission and Formatting Instructions for ICML 2026”
   2. Across the paper, \citep has been commonly used in place of \citet.

[1] Wang, R., Zhou, Z., Zhang, T., Wang, L., Xu, X., Liao, X., & Li, K. (2023). Learning to Branch in Combinatorial Optimization with Graph Pointer Networks. IEEE/CAA Journal of Automatica Sinica, 11, 157-169.

[2] Nguyen Vinh Toan, Nguyen Van Hop. Graph convolutional network with multiple aggregators for learning to branch. Knowledge-Based Systems, Volume 326, 27 September 2025. [[code](https://github.com/inmyrealm-james/MA-GNN-L2B)]

[3] Hao Yuan, Wenli Ouyang, Changwen Zhang, Congrui Li, Yong Sun. OPTFM: A Scalable Multi-View Graph Transformer for Hierarchical Pre-Training in Combinatorial Optimization. NeurIPS 2025. [[OR & code in supplement](https://openreview.net/forum?id=24tuzE5KZc)]

[4] Frederik Wenkel, Semih Cantürk, Stefan Horoi, Michael Perlmutter, Guy Wolf. Towards a General Recipe for Combinatorial Optimization With Multi-Filter GNNs. Learning on Graphs (LoG) 2024. [[code](https://github.com/WenkelF/copt)]

---

> ### Author Rebuttal · Authors · 2026-03-31
>
> We sincerely appreciate your valuable comments. All rebuttal tables are available at https://anonymous.4open.science/r/DAB/tables.pdf
>
> **W1.1 Stronger baselines**
>
> Thank you for providing these valuable baselines. Following your suggestion, we have expanded the comparison to OPTFM [R1], MAGCN [R3], and GCON [R4]. We further include a strong graph transformer (Polynormer [R2]) and a SOTA GNN variant GNN-SSM [R5]. These additions substantially broaden the empirical evaluation beyond BGNN. We kindly refer the reviewer to our response to **Reviewer AP6h (Q3&W)** for full results and discussions.
>
> **W1.2 GNN Limitations**
>
> We would like to respectfully clarify that our point is more specific to the standard bipartite local message-passing in MILP, rather than claiming all GNNs invariably suffer from over-smoothing/squashing. While GCON [R4] is effective with 20+ layers on relatively small graphs, in our large-scale WA datasets (more than 120k nodes), we observe a clear degradation when GCON is stacked too deeply. As shown in Table R6, the performance of GCON peaks at shallow depth (2 layers) on WA and drops substantially at 10 and 15 layers. We also examined GNN-SSM [R5], a SOTA GNN specifically designed to handle over-smoothing/squashing, and MAGCN [R3], a novel GNN tailored for MILP. Table R7 shows that they also tend to saturate at 2 layers. In the following Table R15-R17, we use GCON, GNN-SSM and MAGCN with 2 layers. These results reveal that SOTA GNNs may still be insufficient in handling complicated MILP instances. We will follow your suggestion to add detailed discussion in the revised paper.
>
> **W1.3&Q1 Fairness**
>
> We would like to respectfully clarify that Sec. 4.4 already provides empirical evidence of BGNN's limitations in both depth and long-range dependency analysis. Meanwhile, a two-layer BGNN can already cover 4-hop nodes since each BGNN layer has two half-convolutions. Hence, the weaker performance of BGNN in Table 1 is unlikely to be explained solely by an insufficient local range.
>
> More directly to address your concerns, we evaluate 1) a 2-layer version and 2) a $d=32$ version of our model on the same dataset used in Table 1. Results in Table R9-R10 show that our model maintains stronger performance under the same number of layers and even with fewer parameters.
>
> **W2&Q2 Evidence of expressivity over BGNN**
>
> As mentioned before, Sec. 4.4 provides detailed empirical evidence that BGNN suffers from representation collapse, gradient decay, and weakened long-range influence as depth grows (Fig. 3 and 4), whereas our model remains more stable and retains substantial influence from distant nodes. While this is not a formal theoretical analysis, it does provide concrete empirical evidence that our architecture is better at capturing deeper representations and long-range dependencies in MILP. We also supplement a mechanistic explanation, and kindly refer the reviewer to our response to **Reviewer yC9y (W2)**.
>
> **Q3 Efficiency**
>
> We would like to respectfully clarify that our experiments cover large-scale instances. WA instances used in our experiments go up to 61k variables and 64k constraints, as in Table 3 of Han et al. (2023). In addition, as noted in Appendix A.3, our model can handle large-scale MIPLIB instances up to 550k variables and 1.1M nonzeros on our hardware.
>
> Here we compare our model against a full self-attention variant on the WA dataset. Results in Table R11 show that the default linearized approximation provides a favorable efficiency-accuracy trade-off, with only a minor loss in accuracy but a clear gain in runtime.
>
> **Q4 Hyperparameters in Table 15**
>
> We apologize for a typo in Table 15 (will be corrected). We actually use the same hyperparameters ($k_0=50$, $k_1=0$) for BGNN and our method on CA. The only difference occurs on MIS.
>
> The reason for possibly using different $k_0, k_1, \Delta$ lies in the nature of the PaS framework, which solves a trust region problem by fixing variables according to prediction. In PaS, these hyperparameters directly control the aggressiveness of variable fixing. If the settings are too aggressive, unreliable fixings may hurt solution quality or feasibility; if they are too conservative, the acceleration benefit becomes limited. Intuitively, the appropriate operating point depends strongly on the prediction quality, and hence could be sensitive to different prediction models. Here we additionally conduct a controlled comparison under identical hyperparameter settings, where BGNN is evaluated using the same fixing parameters ($k_0=400$, $k_1=400$) as ours. Results in Table R12 show that BGNN performs worse compared to its specially tuned configuration.
>
> Apollo is less sensitive to these hyperparameters, since the fixing behavior is further adjusted by a short solver probing stage to correct some prediction error, hence is more robust to the prediction accuracy. The same configuration works well for both our method and BGNN.

---

> > ### Author Rebuttal · Reviewer_bqNa · 2026-04-04
> >
> > - **[W1]** I appreciate the extensive additional empirical study, and the clarification re: local range. Re: W1.2, I acknowledge the authors’ point that they refer to GNN applications on bipartite graphs. The results are overall convincing, and compare the proposed method with several of the discussed recent baselines succesfully. The tables claim “fair comparisons” where appropriate — given the empirically-driven nature of the work and the extensiveness of the empirical study, I ask the authors to ensure that (i) the experimental settings are explained thoroughly for each method in the revision, and (ii) a reproducible codebase is provided.
> > - **[W2/Q2]** I appreciate the sensible additional mechanistic study which support the claims made on expressivity. I still think the theoretical backing of the paper (which turns to expressivity as an explanation of methodological superiority over BGNNs) is limited as it is experimental/mechanistic in nature, but the authors have partially addressed this concern.
> > - **[Q3, Q4]** Acknowledged, thank you.
> >
> > I commend the authors for a strong rebuttal. Some minor points may remain, but overall I think this work will be of benefit to the neural CO community and am happy to increase my score to 5 (accept).

---

> > > ### Author Response · Authors · 2026-04-04
> > >
> > > Thank you very much for your careful reading and constructive feedback, which is of great importance in improving the quality of our work. We are glad that our responses address most of your concerns, and we sincerely appreciate your positive reassessment.
> > >
> > > We will incorporate your suggestions in the revision by describing the experimental settings more thoroughly for each method and releasing a reproducible codebase (our main code is already avaiable as noted in the paper). We will also further clarify the theoretical point in the revised paper, consistent with the additional discussion in our rebuttal.
> > >
> > > Thank you again for your support and helpful comments!

---

### Official Review · Reviewer_wRPY · 2026-03-14

**Soundness:** 2
**Presentation:** 3
**Significance:** 2
**Originality:** 1
**Overall Recommendation:** 4
**Confidence:** 4

**Summary:**

This paper proposes a unified dual-attention backbone for MILP, using self-attention to model intra-type interactions and cross-attention to capture variable–constraint dependencies. The method is evaluated on instance-level, element-level, and solving-state-level tasks, and is further supported by mechanism analysis on depth and long-range dependency modeling. The empirical results show consistent improvements over BGNN-based baselines. Overall, the paper is technically solid and empirically well organized, but its novelty and experimental scope still need to be better justified.

**Compliance With Llm Reviewing Policy:**

Affirmed.

**Final Justification:**

The authors have addressed my main concerns. Although the distinction from OPTFM is not large at a high level, the rebuttal clarifies that the proposed design has meaningful differences, and the additional comparisons provide empirical support for its advantage over OPTFM in the reported settings. The authors also add several supplementary experiments to strengthen the empirical evaluation.

**Key Questions For Authors:**

1.	Could the authors expand the empirical comparison to include stronger transformer-style or heterogeneous-attention-style baselines across the main downstream tasks, rather than focusing primarily on BGNN-based comparisons?

2.	What is the intended practical use of the graph-level optimal objective prediction task in realistic MILP workflows? In addition, how does the method handle variation in objective scale across instances of different sizes, and was any normalization applied to support generalization?

3.	For the unfoldable synthetic dataset, how can the authors rule out the possibility that the objective prediction results are partially driven by the relatively restricted coefficient scale in the generated data, rather than by learning genuinely useful global structural information?

**Limitations:**

No. The paper would benefit from a brief discussion of its main limitations.

**Strengths And Weaknesses:**

Strengths:

1.	The paper proposes a unified dual-attention backbone for MILP and demonstrates its effectiveness across multiple downstream tasks.

2.	The experimental study is relatively comprehensive, covering instance-level, element-level, and solving-state-level evaluations.

3.	The mechanism analysis is thorough and provides useful evidence on depth, long-range dependency modeling, and the contribution of each module.

Weaknesses:

1.	The novelty of the proposed backbone appears somewhat limited. Similar hybrid self-/cross-attention ideas have already appeared in recent attention-based optimization models such as OPTFM [1], so the paper should more clearly position its methodological contribution.

2.	Although the paper includes an OOD generalization experiment on MIPLIB for the element-level task, the evaluation of generalization is still not systematic. Given that the proposed method is presented as a general MILP backbone, the paper would be strengthened by a more consistent assessment of cross-dataset generalization across different task levels.

3.	The baseline selection is somewhat limited. Most of the main comparisons are against the BGNN baseline, without systematic evaluation against stronger transformer-style or heterogeneous-attention-style baselines.

4.	The practical value of the graph-level objective prediction task is not sufficiently clear. It is also unclear how the method handles variation in the objective scale across instances of varying sizes.

[1] Yuan, Hao, et al. "OPTFM: A Scalable Multi-View Graph Transformer for Hierarchical Pre-Training in Combinatorial Optimization." The Thirty-ninth Annual Conference on Neural Information Processing Systems.

Presentation Issue:

1.	Figure 3 does not clearly indicate which results correspond to BGNN and which correspond to the proposed method.

---

> ### Author Rebuttal · Authors · 2026-03-31
>
> Thank you for the valuable comments. All rebuttal tables are available at https://anonymous.4open.science/r/DAB/tables.pdf
>
> **W1 Positioning and novelty**
>
> Thank you for pointing out the connection to OPTFM. While both works combine self-attention and cross-attention, our design is significantly different. In particular, OPTFM follows the sequential paradigm of BGNN in that self-attentions are applied to variables/constraints first, and then two cross-attentions are sequentially conducted (v→c→v). In contrast, our self- and cross-attention modules work *in parallel* as in Fig. 1 and Section 3.1, which effectively mitigates the information bottleneck of constraint nodes in the two-stage message passing, and reduces interference between local and global features. This design better reflects the inherent dependency of MILP, where variable–variable, constraint–constraint, and variable–constraint interactions naturally coexist. Empirically, OPTFM is hard to go deep. Table R7 shows that its performance significantly drops beyond 1 layer, while Table R1 shows that our model is still not saturated at 5 layers. In Table R15, we supplement a systematic comparison with OPTFM, highlighting the different behaviors of the two architectures. We will clarify these distinctions and better position our contribution in the revised paper.
>
> **W2 Cross-dataset generalization**
>
> Our primary goal is to design a neural backbone that can support different learning tasks and training distributions, rather than to develop a single pre-trained model for universal cross-distribution generalization. Nevertheless, we agree a more systematic cross-dataset evaluation is valuable. Here we supplement such study on the element-level tasks by directly testing the models trained on WA, MIS, and CA on the three testing sets, forming a 3×3 transfer matrix. We exclude IP due to feature dimensionality mismatch. As in Table R4, direct cross-dataset transfer is challenging for both methods (BGNN even failed on WA->MIS with zero MCC), but our model achieves better average OOD performance (mean off-diagonal MCC: 0.1805 vs 0.1573). Interestingly, for both methods, models trained on CA generalize better, suggesting CA may cover more diverse structural patterns, making it a more transferable training distribution. However, as in Table 9 in A.3, BGNN's CA model almost fails in MIPLIB while our CA model maintains strong performance. We believe this is stronger evidence of generalization since MIPLIB contains real-world instances with much more diverse distributions than the synthetic transfer evaluation in Table R4.
>
> **W3&Q1 Stronger baselines**
>
> BGNN is the primary baseline since it remains the de facto architecture for MILP. Its two-stage message passing scheme is still adopted even in the advanced OPTFM. Nevertheless, we agree comparisons with stronger baselines are important. Table 7 already includes BGNN variants with graph- and self-attention, and an advanced graph transformer (BGT [R2]). Here we further compare with stronger transformer-style (BGT), heterogeneous-attention-style (OPTFM) and advanced GNN models (MAGCN [R3], GCON [R4], GNN-SSM [R5]) . We kindly refer the reviewer to our response to **Reviewer AP6h (W3&Q3)** for detailed results.
>
> **W4&Q2 Objective prediction**
>
> The practical significance of objective prediction lies in providing a global prior without solving the instance. It can be used in various ways such as guiding node selection in B&B, early-stopping solvers, and providing immediate quality estimates for scenario screening/what-if analysis without solving.
>
> Our objective prediction experiments completely follow the open-source code of Chen et al. 2023, where objective coefficients are sampled from N(0, 0.01), keeping objectives within a relatively small and consistent range across instances. Hence the original setup does not apply normalization (and neither do we) and raw objective values are used directly in the loss. While this ensures fair comparison, we agree handling objective variations could be an interesting future work.
>
> **Q3 Scale impact**
>
> Here we add two controls in Table R5: 1) a constant predictor always outputs the training set mean, leading to a test MSE of 1.46e-01 showing the narrow target range alone cannot explain the low prediction error; 2) we train a structure-agnostic XGBoost regressor using only instance-level statistics (size, objective coefficient statistics, and constraint matrix statistics) with no graph message passing, which is only slightly better than the constant baseline. By contrast, structure-aware models (BGNN and ours) performs much better, showing the importance of informative structural signals captured by structure-aware models.
>
> **Fig. 3 Presentation**
>
> As in the caption, (a–d) and (e–h) correspond to BGNN and our method, respectively. We will make this clearer in the revised version.

---

> > ### Author Rebuttal · Reviewer_wRPY · 2026-04-04
> >
> > The authors have addressed my main concerns. Although the distinction from OPTFM is not large at a high level, the rebuttal clarifies that the proposed design has meaningful differences, and the additional comparisons provide empirical support for its advantage over OPTFM in the reported settings.
> >
> > I also appreciate the authors’ effort in adding several supplementary experiments to strengthen the empirical evaluation. Overall, I consider my main concern sufficiently resolved, and I am increasing my score to 4.

---

> > > ### Author Response · Authors · 2026-04-04
> > >
> > > We sincerely appreciate the time and care you devoted to reading our rebuttal and sharing your thoughtful feedback. It is very encouraging to know that our additional clarifications and empirical study were helpful in addressing your main concerns.
> > >
> > > We are also grateful that you recognized the value of the supplementary experiments we conducted to strengthen the evaluation. Your comments have been very helpful for improving the presentation of our work.
> > >
> > > Thank you again for your constructive feedback and kind support.

---

### Official Review · Reviewer_yC9y · 2026-03-16

**Soundness:** 3
**Presentation:** 3
**Significance:** 2
**Originality:** 2
**Overall Recommendation:** 4
**Confidence:** 4

**Summary:**

This paper introduces an attention-driven neural backbone for MILP. By leveraging sparse, it enables efficient global information exchange (self-attention) and variable-constraint interaction (cross-attention), significantly enhancing performance of downstream tasks in combinatorial optimization problems.

**Compliance With Llm Reviewing Policy:**

Affirmed.

**Key Questions For Authors:**

Lacks a detailed exploration of the impact of hyper-paramters like layer depth (T) and hidden dimension (d) on solving efficiency, optimal configurations for large-scale instances remain to be verified.

**Limitations:**

yes

**Strengths And Weaknesses:**

Strengths: leverages MILP sparsity to reduce complexity, explicitly incorporates the MILP structure via coefficient-aware mechanisms, clear presentation;
Weakness: optimal configurations for large-scale instances remain to be verified, and the justification for why attention outperforms GNNs in global interaction within highly constrained spaces is primarily empirical.

---

> ### Author Rebuttal · Authors · 2026-03-31
>
> We greatly appreciate your positive comments. All rebuttal tables are available at https://anonymous.4open.science/r/DAB/tables.pdf
>
>
> **W1&Q1 Impact of layer depth $T$ and hidden dimension $d$ on large-scale instances**
>
> To better understand the effect of $T$ and $d$, we perform a systematic sweep over $T=1,2,3,4,5$ and $d=16,32,64,128,256$ on the WA dataset in element-level prediction, a representative large-scale benchmark (maximum 61000 variables and 64480 constraints as in Table 3 in Han et al., 2023). Results in [Table R1](https://anonymous.4open.science/r/DAB/tables.pdf) show that the model achieves consistently strong performance when $T>2$, indicating robustness to the choice of $T$ and $d$. We also can see a clear trend that increasing $T$ and $d$ generally improves prediction accuracy. However, this also increases inference latency and GPU memory usage as shown in [Tables R2 and R3](https://anonymous.4open.science/r/DAB/tables.pdf). While it is possible to obtain better results with larger $T$ and $d$, we use $T=4$ and $d=64$ as a reasonable trade-off between prediction accuracy and efficiency. This setting also leads to robustly good performance in other datasets.
>
> **W2 Justification of attention in capturing global interaction**
>
> We agree that the current justification is mainly empirical. Here we provide a brief mechanistic explanation. From the structural perspective, MILP instances are represented as bipartite graphs where dependencies between variables are often mediated through multiple constraints. Capturing long-range dependencies between variables with message-passing GNNs (such as BGNN) typically requires stacking multiple layers, which may introduce information bottlenecks during message aggregation. In contrast, the proposed dual-attention mechanism allows nodes to directly attend to a broader set of relevant variables and constraints, enabling more flexible modeling of global constraint-variable dependencies. Moreover, the self-attention and cross-attention modules operate *in parallel* (Fig. 1), rather than following the sequential two-stage propagation (variable->constraint->variable) used in BGNN. This design allows the model to capture richer global interactions while reducing information bottlenecks. We will clarify this design motivation in the final version. Interestingly, [Table R1](https://anonymous.4open.science/r/DAB/tables.pdf) also shows that performance becomes stable after only a few attention layers (around $T=3$–$4$), suggesting that our attention-based architecture can capture global interactions with relatively shallow depth. This behavior is consistent with the intuition that attention enables more direct modeling of long-range dependencies compared with multi-hop message passing.

---

> > ### Author Rebuttal · Reviewer_yC9y · 2026-04-03
> >
> > Thanks for the authors' response, after reading the rebuttal, I think it is appropriate to maintain my current score.

---

> > > ### Author Response · Authors · 2026-04-04
> > >
> > > We are glad that our clarifications have addressed your concerns. We sincerely thank you for your careful reading and valuable suggestions.

---

### Decision · Program_Chairs · 2026-04-30

**Decision:**

Accept (regular)

**Comment:**

The paper presents a good architecture for solving efficiently sparse MILPs. Reviewers mostly agree on the architectural contribution and good results. Especially the additional results and discussions presented in the rebuttal should be added to a final version of the paper. On the downside, reviewers also raised limited novelty and, in the submitted form, evaluation that has room for improvement.
All in all the paper meets the bar for publication and I strongly urge the authors to include the discussion and results form the rebuttal in the published version.